# Accurate Forgetting for Heterogeneous Federated Continual Learning

**Abudukelimu Wuerkaixi**[* 1]    **Sen Cui**[* 1]    **Jingfeng Zhang**[* 2,3]    **Kunda Yan**[1]
**Bo Han**[4,3]    **Gang Niu**[3]    **Lei Fang**[5]    **Changshui Zhang**[† 1]    **Masashi Sugiyama**[† 3,6]

[1] Institute for Artificial Intelligence, Tsinghua University (THUAI)
  Beijing National Research Center for Information Science and Technology (BNRist)
  Department of Automation, Tsinghua University, Beijing, P.R.China
[2] The University of Auckland    [3] RIKEN    [4] Hong Kong Baptist University
[5] DataCanvas Technology Co., Ltd.    [6] The University of Tokyo

## Abstract

Recent years have witnessed a burgeoning interest in federated learning (FL). However, the contexts in which clients engage in sequential learning remain under-explored. Bridging FL and continual learning (CL) gives rise to a challenging practical problem: federated continual learning (FCL). Existing research in FCL primarily focuses on mitigating the catastrophic forgetting issue of continual learning while collaborating with other clients. We argue that forgetting phenomena are not invariably detrimental. In this paper, we consider a more practical and challenging FCL setting characterized by potentially unrelated or even antagonistic data/tasks across different clients. In the FL scenario, statistical heterogeneity and data noise among clients may exhibit spurious correlations which result in biased feature learning. While existing CL strategies focus on the complete utilization of previous knowledge, we found that forgetting biased information was beneficial in our study. Therefore, we propose a new concept *accurate forgetting* (AF) and develop a novel generative-replay method `AF-FCL` that selectively utilizes previous knowledge in federated networks. We employ a probabilistic framework based on a normalizing flow model to quantify the credibility of previous knowledge. Comprehensive experiments affirm the superiority of our method over baselines.

## 1 Introduction

Continual learning is a learning scenario where a model tries to learn a series of new arriving tasks and maintain performance on old tasks (Thrun, 1994; Kumar & Daume III, 2012; Li & Hoiem, 2016; Jeon et al., 2023). This approach, inspired by human lifelong learning, is central to advancing the development of artificial general intelligence. Since birth, a person would gather experience about real world by constantly learning various tasks and remembering them. Humans not only accumulate knowledge through self-directed learning but also collaboratively learn from others. However, concerns about data privacy and communication overhead arise when cooperating with others. Federated learning, which has attracted significant interests and gained various applications in industry (McMahan et al., 2017; Yang et al., 2019; Li et al., 2021), has been an alternative to addressing these concerns. This leads to the concept of federated continual learning (FCL) (Qi et al., 2023), incorporating continual learning into federated learning.

In FCL, the goal is that clients learn models for their private sequential tasks collaboratively without violating the data privacy of individual clients. This could encounter challenges from three fronts. One is *statistical heterogeneity* due to non-IID data across local clients. Such heterogeneity could severely degrade performance (Qu et al., 2022) when learning from clients collaboratively. Another is catastrophic forgetting, stemming from restricted access to data from previous tasks due to realistic factors such as storage constraints, privacy issues, etc (Wang et al., 2023). This can lead the model to lose its ability to perform previous tasks proficiently after assimilating new tasks. There are

---

[*]These authors contributed equally to this work.
[†]Corresponding authors
  Code is at: https://github.com/zaocan666/AF-FCL.

a few studies seeking to address the above two problems in FCL. For example, Usmanova et al. (2021) extended the *Learning without Forgetting* (Li & Hoiem, 2016) method to the FCL scenario, memorizing previous tasks among all clients. The third concern is associated with the potential introduction of feature bias resulting from the federated scenario, which in turn could impact the memory within CL models. Research indicates that the memorization of noisy labels can significantly impair the model's performance (Han et al., 2020).

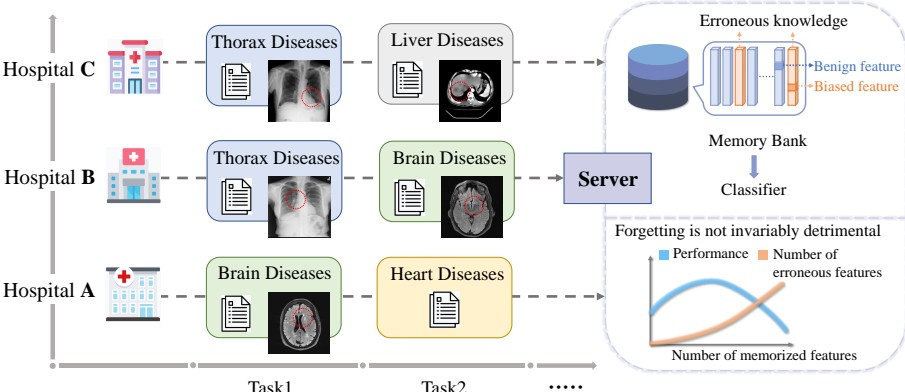

Figure 1: Illustration of the FCL problem. Multiple hospitals within a federated learning network engage in the sequential acquisition of disease prediction tasks. The global memory bank, a crucial tool for the classifier in mitigating catastrophic forgetting, may possess biased features arising from statistical heterogeneity. Notably, the overall performance of the classifier could suffer degradation without strategic forgetting (The experimental verification is in Sec. 4).

Existing research developing FCL methods mainly assumed that thorough memorization of previous tasks yields overall performance benefits (Usmanova et al., 2021; Qi et al., 2023). Elaborate strategies were employed to memorize previous information (Yoon et al., 2021a; Liu et al., 2023). In practice, feature bias typically exists in the dataset, especially when there are lots of clients within the federated network. Because of such statistical heterogeneity, biased or even harmful information from particular clients may reside in the *memory bank* (i.e., memory buffer, generative models or model parameters) as shown in Figure 1. The federated model may inadvertently learn to identify and rely upon spurious correlations arising from diverse tasks among multiple clients. Furthermore, the model may integrate label noise (Zhang et al., 2024) introduced by a few clients. For example, in a federated learning system implemented among hospitals nationwide, these medical institutions may encounter varying disease profiles over time. Besides, hospitals located at distinct geographical areas often cater to diverse distributions as depicted in Figure 1. Therefore, strategically mitigating erroneous knowledge during the acquisition of new tasks is required.

Motivated by the phenomenon in reality that the new arriving tasks of each client may not be correlated, we consider a more practical and challenging FCL setting in this paper: *limitless task pool* (LTP). From a temporal perspective, the tasks that a single client randomly selects from the LTP at various time points might be unrelated or even antagonistic, thereby presenting a significant challenge for model learning. To overcome the problem, we propose a novel generation-based method *Accurate Forgetting Federated Continual Learning* (AF-FCL). We argue that the forgetting phenomena are not invariably detrimental (Han et al., 2020). Conversely, accurate forgetting mitigates the negative impact of the heterogeneity on model learning.

Instead of learning a generative adversarial network (GAN) for indiscriminate generative-replay in existing FCL methods (Qi et al., 2023), AF-FCL aims to facilitate a selective utilization of previous knowledge through *correlation estimation*. In order to accurately identify benign knowledge from previous tasks, we achieve correlation estimation with a learned normalizing flow (NF) model (Durkan et al., 2019; Winkler et al., 2019; Rezende & Mohamed, 2015) in feature space. Specifically, an NF model could map an arbitrarily complex data distribution to a pre-defined distribution through a sequence of bijective transformations. Such invertability enables the NF to have a lossless memory of the input knowledge and accurately estimate the probability density of observed data. While the information in the NF model could contain biased features or spurious correlation due to heterogeneous data, we suggest outlier features with respect to the current tasks are suspicious and may pose a threat

to the learning process. More precisely, the credibility of a particular feature could be quantified with its probability density in the current tasks.

Experimental results corroborate that `AF-FCL` significantly outperforms all baselines on a series of benchmark datasets. We summarize our key contributions as follows:

- We consider a more practical and challenging FCL setting. We suggest the harm of remembering biased or irrelevant features, which could be unavoidable in the federated scenario due to statistical heterogeneity.

- We propose the concept *accurate forgetting* and develop a novel generative method, `AF-FCL`. It adaptively mitigates erroneous information by correlation estimation with an NF model.

- We conduct extensive experiments on a series of benchmark datasets. The results with ablation studies demonstrate the effectiveness and superiority of our proposed accurate forgetting over existing state-of-the-art methods.

## 2 RELATED WORK

**Continual Learning**. Continual learning has witnessed the development of diverse methodologies (Lange et al., 2022), which can be roughly divided into three families: (I) Regularization-based methods: LwF employs the knowledge distillation loss, where the previous model's output is utilized as soft labels for the current tasks when working with new data (Li & Hoiem, 2016). Stable SGD (Mirzadeh et al., 2020) demonstrated performance enhancements by calibrating pivotal hyper-parameters and systematically reducing the learning rate upon the arrival of each task. (II) Parameter isolation methods: Rusu et al. (2016) suggested augmentation of the model with new branches tailored to incoming tasks. (III) Replay-based methods: Generative replay-based methods use an auxiliary generator to model the data distribution of acquired knowledge, producing synthetic data for replay in instances (Odena et al., 2017; Wu et al., 2018). While existing research predominantly focused on the efficient memorization of past knowledge, we turn our attention to a more foundational question: *is prior knowledge perpetually beneficial?*

**Federated Learning**. Federated learning represents a distributed learning paradigm among multiple clients and a central server. Researchers have been endeavoring to address the statistical heterogeneity by developing a comprehensive global model (Wang et al., 2020). Mohri et al. (2019) aimed to achieve a fair distribution of model performance by optimizing its efficacy across any given target distribution. Zhu et al. (2021b) suggested the utilization of a generator to aggregate user information. This, in turn, guides the local training by employing the acquired knowledge as an inductive bias. In this work, we consider a more challenging learning problem associated with statistical heterogeneity in federated scenarios: how to facilitate collaboration when all clients are tackling different tasks?

**Federated Continual Learning**. To date, there are a few studies in the domain of federated continual learning. Casado et al. (2020) studied the scenario of data distributions changing over time in federated learning. Federated reconnaissance presented a scenario with incrementally new classes during training and proposed to utilize prototype networks (Hendryx et al., 2021). Guo et al. (2021) proposed a regularization-based algorithm and a new theoretical framework for it. Usmanova et al. (2021) presented a distillation-based method to deal with catastrophic forgetting, using previous model and global model as teachers for the training of local models. Yoon et al. (2021b) proposed a novel parameter isolation method for the federated diagram, where the network weights are decomposed into global parameters and task-specific parameters. Dong et al. (2022) considered a federated class-incremental setting and developed a distillation-based method to alleviate catastrophic forgetting from both local and global perspectives. Qi et al. (2023) customized the generative replay based method ACGAN with model consolidation and consistency enforcement. Our method considers the issue of memorizing biased feature due to statistical heterogeneity, exhibiting notable differences compared to the aforementioned methods.

## 3 PROBLEM DEFINITION

### 3.1 NOTATIONS

**Continual Learning.** In standard continual learning scenario, there are a sequence of tasks $\mathcal{T} = \{\mathcal{T}^1, \mathcal{T}^2, \ldots, \mathcal{T}^T\}$, where $T$ is the number of tasks, and $\mathcal{T}^t$ is the $t$-th task. Each dataset is composed

of $n^t$ pairs of data and labels: $D^t = \{x_k^t, y_k^t\}_{k=1}^{n^t}$. When learning on the $t$-th task, one has no direct access to previous data $D^{t'}, t' < t$. The goal of continual learning is to effectively manage the current task while preserving its performance on all previous tasks:

$$\min_{\theta^t} \; [\mathcal{L}(\theta^t; \mathcal{T}^1), \mathcal{L}(\theta^t; \mathcal{T}^2), \ldots, \mathcal{L}(\theta^t; \mathcal{T}^t)], \tag{1}$$

where $\mathcal{L}$ is the risk objective of tasks and $\theta^t$ is the model parameters learned on the $t$-th task.

**Federated Learning and Statistical Heterogeneity.** In federated learning scenario, there are $N$ clients, and each client owns a private dataset. The goal of federated learning is collaboratively learning models without accessing the datasets belonging to the local clients. The data of clients consists of the input space $X_i$ and output space $Y_i$, where $X_i$ and $Y_i$ are shared across all clients. There are $n_i$ samples in the $i$-th client denoted as $\{x_k^i, y_k^i\}_{k=1}^{n_i}$. Different clients may exhibit non-identical joint distributions $p(x, y)$ of features and labels, i.e., $p(x_{i_1}, y_{i_1}) \neq p(x_{i_2}, y_{i_2})$, where $i_1 \neq i_2$.

## 3.2 Federated Continual Learning

FCL refers to a practical learning scenario that melds the principles of federated learning and continual learning. Suppose there are $N$ clients, and each client possesses a private series of datasets $\{D_k^t\}_{t=1}^T$. Please note that, at a given step $t$, client $k$ can only have access to $D_k^t$ as in continual learning. In existing literature, the primary focus is on a specific task reshuffling setting, wherein the task set is identical for all users, yet the arrival sequence of tasks differs (Yoon et al., 2021a). In practical scenarios, it may be observed that the task set of clients is not necessarily correlated. Thus we consider a practical setting, the limitless task pool (LTP), denoted as $\mathcal{T}$. For each client, the dataset $D_k^t$ of the $k$-th client at step $t$ corresponds to a particular learning task $\mathcal{T}_k^t \subset \mathcal{T}$. There is no guaranteed relation among the tasks $\{\mathcal{T}_k^1, \mathcal{T}_k^2, \ldots, \mathcal{T}_k^T\}$ in the $k$-th client at different steps. Similarly, at step $t$, there could be no relation among the tasks $\{\mathcal{T}_1^t, \mathcal{T}_2^t, \ldots, \mathcal{T}_N^t\}$ across different clients.

**Limitless Task Pool.** In the setting of LTP, tasks are selected randomly from a substantial repository of tasks, creating a situation where two clients may not share any common tasks, i.e., $\left| \{\mathcal{T}_p^i\}_{i=1}^{t_p} \cap \{\mathcal{T}_q^i\}_{i=1}^{t_q} \right| \geq 0, \; p, q = 1, 2 \ldots, N$. More importantly, clients possess diverse joint distributions of data and labels $p(x, y)$ due to statistical heterogeneity. Therefore, features learned from other clients could invariably introduce bias when applied to the current task.

**Biased Features.** The bias originating from a particular client can adversely affect the performance of the model across different clients and a range of tasks. We tackle a more practical and challenging FCL problem that differs from the task reshuffling setting (Yoon et al., 2021a) from two perspectives: (I) For different steps, tasks allocated to each client are randomly drawn from an extensive task pool. (II) For different clients, tasks across various clients may be unrelated or even contradictory in each step, consequently amplifying bias during the learning process.

Our goal is to facilitate the collaborative construction of the global model with parameters $\theta$. Under the privacy constraint inherent in federated learning and continual learning, we aim to harmoniously learn current tasks while preserving performance on previous tasks for all clients, thereby seeking to optimize performance across all tasks seen so far by all clients, i.e.,

$$\min_{\theta^t} \; [\mathcal{S}_1^L, \mathcal{S}_2^L, \ldots \mathcal{S}_N^L], \text{where } S_i^L = [\mathcal{L}(\theta^t; \mathcal{T}_i^1), \mathcal{L}(\theta^t; \mathcal{T}_i^2), \ldots, \mathcal{L}(\theta^t; \mathcal{T}_i^t)]. \tag{2}$$

# 4 Validation of Accurate Forgetting

In this section, we present the results on a noisy dataset to intuitively demonstrate the effectiveness of our motivation and approach.

## 4.1 Dataset with Label Noise

We argue that forgetting is not invariably detrimental within the realm of FCL and propose the concept of accurate forgetting. To validate our argument and the efficacy of our proposed method, we curate the EMNIST-noisy dataset, wherein a subset of noisy clients is simulated by introducing random labels to the data. Additionally, we acknowledge the presence of noise in practical datasets, notably in the form of label noise.

As a character image dataset, the EMNIST-noisy dataset comprises 8 clients, each encompassing 6 tasks, with each task containing 2 classes of character images. We randomly select several clients and

assign random labels for their initial three tasks, as displayed in Figure 2(a). These incorrect labels have the potential to propagate adverse effects, affecting subsequent task learning across different clients through the memory bank. After learning sequentially on all tasks, we evaluate the final three tasks, which do not contain any noisy labels. This evaluation allows us to exclusively assess the impact of incorporating noisy information from previous tasks into the memory bank.

## 4.2 RESULTS

The baselines in Figure 2(b) are representative CL and FCL methods. It is observed that: (I) the performance of the baselines demonstrates inferiority compared to the naive FedAvg method; (II) the performance of the baselines suffers a rapid deterioration with an increasing number of noisy clients.

These CL and FCL baseline methods are meticulously designed to effectively retain knowledge from previous tasks. However, the presence of noisy clients introduces harmful information into the model learning process. The memorization of such erroneous information proves detrimental to the overall performance. Consequently, the baselines exhibit suboptimal performance compared to FL method, which does not employ explicit memorization techniques. In contrast, our approach incorporates adaptive mechanisms to mitigate the impact of erroneous information. By effectively alleviating the adverse influence of noisy clients, our method consistently surpasses all baselines. Notably, the performance of our method maintains relative stability even with an increasing number of noisy clients in the dataset.

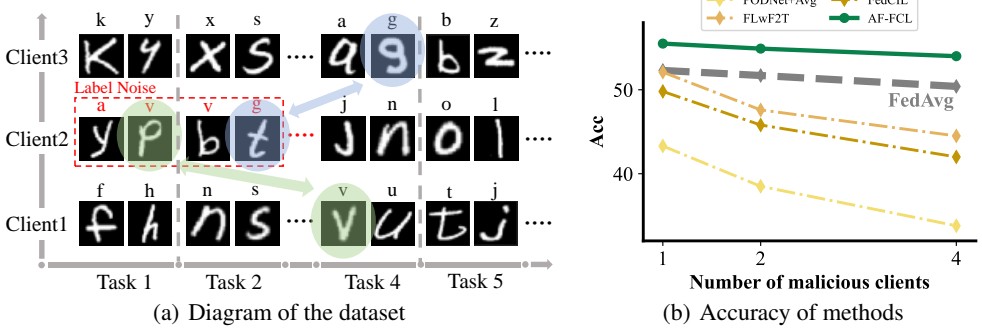

(a) Diagram of the dataset  (b) Accuracy of methods

Figure 2: Illustration of the EMNIST-noisy dataset and results. (a) The initial several tasks in Client2 exhibit label noise. (b) The average accuracy of methods is presented with respect to an increasing number of malicious clients. The baseline methods are illustrated by dash-dotted lines, while our method is depicted with solid line.

## 5 METHODOLOGY

### 5.1 PRELIMINARY: NORMALIZING FLOW

Normalizing flow is a type of generative model. It is able to map a complex, multi-modal distribution to a simple probability distribution such as standard Gaussian distribution through a sequence of smooth and invertible transformations (Rezende & Mohamed, 2015). In particular, an NF model is a diffeomorphism $g$ composed of a series of invertible transformations $g = g_1 \circ g_2 \ldots \circ g_k$, of which a widely applied transformation is affine coupling layer (Kingma & Dhariwal, 2018).

**Lossless Memory.** Through meticulous design of the invertible layers, normalizing flow accomplishes a bijective transformation, preserving the one-to-one correspondence between the elements of the input and output spaces. The bijectivity ensures a lossless memory of the original input. Consequently, this inherent property of NF is pivotal in enabling the accurate modeling of complex distributions, and stands central in generative applications.

**Exact Likelihood Estimation.** The invertibility enables precise estimation of the probability density of data samples within the learned dataset. Specifically, with a target dataset $Z = \{z_i\}_{i=1}^{n}, z_i \in \mathbb{R}^d$ and a prior distribution $p_u(u), u \in \mathbb{R}^d$, an NF model learns the diffeomorphism $g$ with the parameters $\phi$ that maps dataset distribution $p_z$ to the prior: $u = g(z)$. Under above transformation, the probability density of the given datapoint $z$ can be computed as:

$$\log p_z(z) = \log p_u(u) + \log \left| \det \frac{\partial u}{\partial z} \right| = \log p_u(g(z)) + \sum_{l=1}^{k-1} \log \left| \det \frac{\partial g^{l+1}}{\partial g^l} \right|, \qquad (3)$$

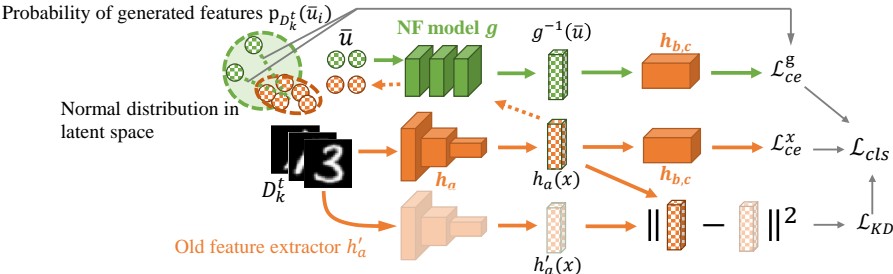

Figure 3: The diagram of training the classifier locally with our method. The training objective consists of three integral components: (I) $\mathcal{L}_{ce}^{g}$, representing the objective for training using features generated and estimated probabilities derived from the global NF model; (II) $L_{ce}^{x}$, corresponding to the objective for training using original data; (III) $L_{KD}$, which denotes the objective for knowledge distillation within the feature space.

where $g^l$ denotes input of the $l$-th transformation of NF model. The transformations of the NF model are deliberately crafted to facilitate efficient computation of their Jacobian determinants $\left|\det \frac{\partial g^{l+1}}{\partial g^l}\right|$. A conditional NF model can take label $y$ as conditional information in likelihood estimation $p_z(z, y)$.

**The Training of NF Models.** The training objective of NF model is also derived from Eq. 3 , trained to maximize the likelihood of samples from target dataset $Z$, i.e.,

$$\mathcal{L}_{NF}(g; Z) = -\frac{1}{n}\sum_{i=1}^{n} \log p_z(z_i) = -\frac{1}{n}\sum_{i=1}^{n}\left(\log p_u(g(z_i)) + \sum_{l=1}^{k-1}\log\left|\det\frac{\partial g_i^{l+1}}{\partial g_i^l}\right|\right). \quad (4)$$

## 5.2 An Overview of AF-FCL

In FCL, statistical heterogeneity among clients brings extra challenges for continuously learning a sequence of tasks. Especially in LTP setting, particular clients could possess unrelated tasks and biased dataset. When bias or spurious correlation from particular clients is memorized by the model, a decline in model performance may occur in the task sequences of all clients. Therefore, a direct deployment of continual learning methods designed to mitigate catastrophic forgetting is hard to address the heterogeneity issues in FCL.

We propose a novel method `AF-FCL` , which adaptively utilizes memorized knowledge and learns unbiased feature for all clients under the FedAvg framework (McMahan et al., 2017). The training schematic of the classifier in each client is illustrated in Figure 3. Overall, the implementation of `AF-FCL` consists of the following components: (I) *feature generative-replay*. To prevent complete forgetting, we train a global NF model in the feature space of classifier for generative replay. (II) *knowledge distillation*. Additionally, we employ knowledge distillation in the feature space to mitigate significant drift, thereby enhancing the stability of the training process for the NF model. (III) *correlation estimation*. We suggest that features exhibiting outlier characteristics with respect to the current tasks can potentially undermine the learning process. Therefore, we assess the reliability of the generated feature by its probability density within the current tasks.

## 5.3 Accurate Forgetting for heterogeneous FCL

The above Sec.5.2 gives an overview of our method. In this section, we provide a detailed description of `AF-FCL` and how it is implemented.

**Generative-replay in Feature Space.** We consider the classification tasks, where we need to train a classifier with $L$ layers: $h = \{h_1, h_2, \ldots, h_L\}$. We split the classifier into three sub-modules: $h_a = \{h_1, h_2, \ldots, h_l\}$, $h_b = \{h_{l+1}, \ldots, h_{L-1}\}$, $h_c = \{h_L\}$. The $h_a$ and $h_b$ are two successive feature extractors, $h_c$ is the classifier head. To maintain the performance on previous tasks, we train a conditional normalizing flow model $g$ in the feature space, which is the output space of $h_a$. In this way, the normalizing flow model retains the feature of previous tasks. The NF model $g$ is trained globally with FedAvg algorithm using client datasets and sampled data:

$$\widetilde{\mathcal{L}}_{NF}(g; D_k^t, G_z) = -\frac{1}{|D_k^t|}\sum_{x_i, y_i \sim D_k^t}\log p_z(h_a(x_i), y_i) - \frac{1}{|G_z|}\sum_{z_i, y_i \sim G_z}\log p_z(z_i, y_i), \quad (5)$$

where $D_k^t$ is the dataset of the $t$-th task in the $k$-th client, and $p_z$ is the likelihood calculated as in Eq. 3. $G_z$ is the feature set sampled from NF model $g'$ ($g'$ is the stored NF model after training on the last task), so that the current NF model avoids forgetting previous features.

Normalizing flows operate within a latent space that maintains dimensional parity with the target data space. Training the NF model in high-dimensional data space $X$ could be computationally intensive. Furthermore, the inherent sparsity of raw data can hinder the NF model's capacity to obtain a representative sample of the data distribution (Brehmer & Cranmer, 2020). Therefore, we train the NF model in the compact, low-dimensional feature space as opposed to the data space, thereby reducing the complexity of generation.

We also leverage the feature space to extract more robust semantic information.

**Knowledge Distillation for a Consistent Feature Distribution.** The NF model is trained in the feature space of classifier to maintain previous knowledge. The NF model retains knowledge from previous tasks, conveying it to the classifier via feature generation. Yet, feature extractor of the classifier undergoes continual modifications throughout the training process. If the feature space of the classifier drifts significantly, the knowledge memorized by the NF model may become obsolete.

Therefore, the feature space of the classifier needs to retain relative consistency during the training. We propose to apply knowledge distillation in the feature space of the classifier to control the drift of feature distribution:

$$\mathcal{L}_{KD}(h; D_k^t) = \frac{1}{n_k^t} \sum_{i=1}^{n_k^t} ||h_a(x_i) - h_a'(x_i)||^2, \tag{6}$$

where $h_a'$ is the stored classifier feature extractor after training on the last task.

**Correlation Estimation for Accurate Forgetting.** From the above, we train the classifier with the aid of NF model by generating features. However, utilizing previous knowledge without discrimination may lead to biased model as stated before. Thus we propose to accurately exploit the memorized knowledge with the characteristics of the NF model for correlation estimation. In particular, when training the classifier for the $t$-th task of client $k$, we firstly map the feature of local data to the latent space of normalizing flow, i.e., $\hat{U}_k^t = \{u_i = g(h_a(x_i))\}_{i=1}^{n_k^t}, \ x_i \in D_k^t$. As the NF models transform the features to a disentangled latent space, which is the centered isotropic multivariate Gaussian. Therefore, we approximate the true distribution $U_k^t$ in each class as a multivariate Gaussian with a diagonal covariance structure. The mean vector $\mu_k^t$ and covariance matrix $\Sigma_k^t$ of $\hat{U}_k^t$ can be easily computed by

$$\mu_k^t = \frac{1}{n_k^t} \sum_{i=1}^{n_k^t} u_i, \quad \Sigma_k^t = \frac{1}{n_k^t} \sum_{i=1}^{n_k^t} \text{diag}(u_i - \mu_k^t) \cdot \text{diag}(u_i - \mu_k^t), \ u_i \in \hat{U}_k^t, \tag{7}$$

where $\text{diag}(u)$ turns the vector $u$ into a diagonal matrix.

For generative replay, we sample a batch of latent vectors in the NF model and project them to feature space: $\bar{U}_g = \{\bar{u}_i, \ \bar{z}_i = g^{-1}(\bar{u}_i), \ \bar{y}_i\}_{i=1}^n, \ \bar{u}_i \in p_u$. Please note that we use bar superscripts to denote generated data. The generated features from NF model represent the knowledge of previous tasks among all clients. However, in FCL scenario, there may exist irrelevant or even biased feature from other clients due to statistical heterogeneity. Enhancing the memorizing of biased feature could cause subpar performance or even failing to converge. Considering that outlier features with respect to the current tasks could be unreliable, we quantify the credibility of generated feature with its relevance to local dataset. To evaluate the correlation between the generated feature and the current task, we propose to use the the probability density of the sampled latent vector $\bar{u}_i$ within the current feature distribution quantified in Eq.(7), i.e.,

$$p_{D_k^t}(\bar{u}_i) = \frac{1}{\sqrt{(2\pi)^d |\Sigma_k^t|}} \exp\left(-\frac{1}{2}(\bar{u}_i - \mu_k^t)^T (\Sigma_k^t)^{-1} (\bar{u}_i - \mu_k^t)\right) \tag{8}$$

The probability $p_{D_k^t}(\bar{u}_i)$ above quantifies the degree of correlation between the current task in the local client and the sampled features from NF model. We use the correlation probability of the generated features to re-weight the loss objective $\mathcal{L}_{ce}^g$. And the final objective $\mathcal{L}_{cls}$ consisting of three

terms is as follows:

$$\mathcal{L}_{ce}^x(h; D_k^t) = \frac{1}{n_k^t} \sum_{i=1}^{n_k^t} \mathcal{L}_{ce}(h(x_i), y_i),$$

$$\mathcal{L}_{ce}^g(h; \bar{U}_g) = \frac{1}{n} \sum_{\bar{u}_i \in \bar{U}_g} p_{D_k^t}(\bar{u}_i) \mathcal{L}_{ce}(h_{b,c}(g^{-1}(\bar{u}_i)), \bar{y}_i), \qquad (9)$$

$$\mathcal{L}_{cls}(h; D_k^t, \bar{U}_g) = \mathcal{L}_{ce}^x(h; D_k^t) + \mathcal{L}_{ce}^g(h; \bar{U}_g) + \mathcal{L}_{KD}(h; D_k^t)$$

where $\mathcal{L}_{ce}^x(h; D_k^t)$ denotes the cross-entropy loss of raw dataset, and $\mathcal{L}_{ce}^g(h; \bar{U}_g)$ denotes the unbiased objective of generated data. With the proposed method, the classifier learns beneficial features from previous tasks and accurately forgetting biased features. Moreover, the NF model memorizes more benign features. Both the NF model and the classifier are expected to be of increasing generalizability with the advancement of training progress. The implementation of `AF-FCL` is in Algorithm 1.

# 6 EXPERIMENTS

## 6.1 EXPERIMENTAL SETTINGS AND EVALUATIONS

**Datasets and Settings.** We curate three FCL datasets with different settings. We use $N$ to denote the number of clients, $T$ to denote the number of tasks in each client, $C$ to denote the number of classes in each task. For the EMNIST-based dataset containing 26 classes of handwritten letter images (Cohen et al., 2017), we set the following two settings with $N$=8, $T$=6, $C$=2. **1) EMNIST-LTP:** in LTP setting, we randomly sampled classes from the entire dataset for each client. **2) EMNIST-shuffle:** in conventional shuffle setting, the task sets are consistent across all clients, while arranged in different orders. **3) CIFAR100**: We randomly sample 20 classes among 100 classes of CIFAR100 (Krizhevsky et al., 2009) as a task for each of the 10 clients, and there are 4 tasks for each client ($N = 10, T = 4, C = 20$). **4) MNIST-SVHN-F**: We set 10 clients with this mixed dataset. Each client contains 6 tasks, and each task has 3 classes.

**Metrics.** We use the metrics of accuracy and average forgetting for evaluation following recent works (Mirzadeh et al., 2021; Yoon et al., 2021a). **Average forgetting** assesses the extend of backward transfer during continual learning, quantified as the disparity between the peak accuracy and the ending accuracy of each task.

## 6.2 BASELINES

We compare our method `AF-FCL` with baselines from FL, CL and FCL. In FL, we consider two representative models **FedAvg** (McMahan et al., 2017) and **FedProx** (Li et al., 2020). In CL, **PODNet** incorporates a spatial-based distillation loss onto the feature maps of the classifier (Douillard et al., 2020). **ACGAN-Replay** employs a GAN-based generative replay method (Wu et al., 2018). The CL models are respectively combined with the FL models. In FCL, **FLwF2T** leverages the concept of knowledge distillation within the framework of federated learning (Usmanova et al., 2021). **FedCIL** extends the ACGAN-Replay method within the federated scenario (Qi et al., 2023). **GLFC** exploits a distillation-based method to alleviate the issue of catastrophic forgetting from both local and global perspectives (Dong et al., 2022).

Table 1: Average accuracy and forgetting on EMNIST-LTP and EMNIST-shuffle dataset.

| Model | EMNIST-LTP | | EMNIST-shuffle | |
|---|---|---|---|---|
| | Accuracy↑ | Forgetting↓ | Accuracy↑ | Forgetting↓ |
| FedAvg | $32.5_{\pm 0.9}$ | $20.8_{\pm 0.8}$ | $70.3_{\pm 0.4}$ | $4.9_{\pm 0.6}$ |
| FedProx | $35.3_{\pm 0.5}$ | $19.2_{\pm 0.6}$ | $69.4_{\pm 0.9}$ | $6.0_{\pm 1.3}$ |
| PODNet+FedAvg | $36.9_{\pm 1.3}$ | $19.8_{\pm 0.9}$ | $71.0_{\pm 0.4}$ | $\mathbf{3.9}_{\pm 0.4}$ |
| PODNet+FedProx | $40.4_{\pm 0.4}$ | $14.3_{\pm 0.5}$ | $70.6_{\pm 0.7}$ | $9.6_{\pm 0.3}$ |
| ACGAN-Replay+FedAvg | $38.4_{\pm 0.2}$ | $9.8_{\pm 0.8}$ | $70.0_{\pm 0.5}$ | $4.7_{\pm 0.3}$ |
| ACGAN-Replay+FedProx | $41.3_{\pm 0.9}$ | $10.4_{\pm 0.7}$ | $70.3_{\pm 1.2}$ | $6.1_{\pm 2.0}$ |
| FLwF2T | $40.1_{\pm 0.3}$ | $15.5_{\pm 0.5}$ | $71.0_{\pm 0.9}$ | $8.1_{\pm 0.8}$ |
| FedCIL | $42.0_{\pm 0.6}$ | $12.4_{\pm 0.3}$ | $71.1_{\pm 0.4}$ | $6.4_{\pm 0.2}$ |
| GLFC | $40.1_{\pm 0.8}$ | $14.3_{\pm 0.5}$ | $74.9_{\pm 0.6}$ | $5.6_{\pm 0.7}$ |
| `AF-FCL` | $\mathbf{47.5}_{\pm 0.3}$ | $\mathbf{7.9}_{\pm 0.5}$ | $\mathbf{75.8}_{\pm 0.2}$ | $4.2_{\pm 0.1}$ |

### 6.3 Experiments on EMNIST-based Datasets

**EMNIST-LTP**. In this dataset, clients may encompass unrelated tasks, thus rendering the dataset challenging. As the results shown in Table 1, some of the CL methods integrated with FL algorithms demonstrate comparable performance to that of FCL methods in the EMNIST-LTP dataset. For instance, the average accuracy of ACGAN-Replay+FedProx is 41.3%, higher than two FCL methods FLwF2T and GLFC. This phenomenon can be attributed to challenge posed by the elevated degree of heterogeneity under the LTP setting, which is difficult for these FCL methods to deal with, consequently diminishing their inherent advantages. Nevertheless, our method outperforms all the baselines in the EMNIST-LTP dataset. We argue that statistical heterogeneity in federated networks inevitably results in biased information residing in the memory bank. Both CL methods and existing FCL methods assume that memorization is beneficial, potentially losing their advantages under LTP setting. Our method adopts accurate forgetting to mitigate the negative impact of heterogeneity and selectively encourages the forgetting of malign information. It shows the highest accuracy rate and lowest forgetting rate.

**EMNIST-shuffle.** Different from the EMNIST-LTP dataset, EMNIST-shuffle represents a more tractable dataset within the conventional setting, resulting in higher overall accuracy rates as in Table 1. The FCL methods exhibit superior accuracy compared to CL methods, underscoring their strength. And our method still showcases a superior capacity than all baselines in this commonly adopted dataset setting.

### 6.4 Experiments on More Complicated Datasets

CIFAR100 comprises 100 classes of images. The composite dataset MNIST-SVHN-F comprises two distinct digit classification datasets: MNIST and SVHN, characterized by complex colors and backgrounds, along with a clothing image classification dataset. Table 2 displays the results of these two challenging datasets CIFAR100 and MNIST-SVHN-F. Different tasks exhibit reliance on varying features. For instance, shape features pertinent to digits differ significantly from those relevant to clothing classification. A naive collaboration among clients may lead to a model overly reliant on spurious correlations, overlooking the importance of task-specific features. We suggest a strategy of selective utilization and memorization of learned feature. By relying on the generated features with a higher correlation, `AF-FCL` significantly exceeds the performance of baselines.

Table 2: Average accuracy and forgetting on CIFAR100 and MNIST-SVHN-F dataset.

| Model | CIFAR100 | | MNIST-SVHN-F | |
|---|---|---|---|---|
| | Accuracy↑ | Forgetting↓ | Accuracy↑ | Forgetting↓ |
| FedAvg | $26.3_{\pm2.5}$ | $8.4_{\pm1.2}$ | $55.7_{\pm1.4}$ | $21.9_{\pm0.9}$ |
| FedProx | $28.7_{\pm1.4}$ | $8.2_{\pm1.0}$ | $56.1_{\pm1.0}$ | $21.3_{\pm1.8}$ |
| PODNet+FedAvg | $30.5_{\pm0.8}$ | $8.6_{\pm1.7}$ | $54.2_{\pm0.8}$ | $20.6_{\pm1.5}$ |
| PODNet+FedProx | $32.5_{\pm0.5}$ | $6.4_{\pm0.4}$ | $56.4_{\pm0.4}$ | $20.0_{\pm1.2}$ |
| ACGAN-Replay+FedAvg | $32.1_{\pm1.6}$ | $5.4_{\pm1.1}$ | $56.0_{\pm0.7}$ | $21.4_{\pm0.8}$ |
| ACGAN-Replay+FedProx | $31.8_{\pm0.7}$ | $6.2_{\pm1.2}$ | $56.4_{\pm2.1}$ | $22.1_{\pm1.4}$ |
| FLwF2T | $30.2_{\pm0.7}$ | $7.2_{\pm1.8}$ | $54.2_{\pm0.6}$ | $25.6_{\pm0.5}$ |
| FedCIL | $33.5_{\pm0.7}$ | $6.5_{\pm1.0}$ | $57.2_{\pm1.7}$ | $19.7_{\pm1.0}$ |
| GLFC | $35.6_{\pm0.6}$ | $6.2_{\pm0.7}$ | $61.8_{\pm0.8}$ | $10.8_{\pm1.3}$ |
| `AF-FCL` | $\mathbf{36.3}_{\pm0.4}$ | $\mathbf{4.9}_{\pm0.1}$ | $\mathbf{68.1}_{\pm0.9}$ | $\mathbf{7.5}_{\pm1.0}$ |

## 7 Conclusion

In this study, we navigate the challenges of continual learning in real-world federated contexts, specifically when faced with data or task streams that might be biased or noisy across clients. Current research in continual learning emphasizes the adverse consequences of "catastrophic forgetting". However, we advocate for a perspective that reveals the merit of selective forgetting, especially as a mechanism to mitigate the biased information induced by statistical heterogeneity in reality. Inspired by it, we present a generative framework, termed as `AF-FCL`, meticulously crafted to achieve targeted forgetting by re-weighting generated features based on inferred correlations. The experimental results clearly demonstrate its effectiveness.

## 8 ACKNOWLEDGMENTS*

This work is funded by the Natural Science Fundation of China(NSFC. No. 62176132) and the Guoqiang Institute of Tsinghua University, with Grant No. 2020GQG0005. MS was supported by JST CREST Grant Number JPMJCR18A2 and a grant from Apple, Inc. Any views, opinions, findings, and conclusions or recommendations expressed in this material are those of the authors and should not be interpreted as reflecting the views, policies or position, either expressed or implied, of Apple Inc. BH was supported by the NSFC General Program No. 62376235 and CCF-Baidu Open Fund.

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

## A DATASETS

We construct a series of datasets comprising multiple federated clients, with each client possessing a sequence of tasks. Suppose we use $N$ to denote the number of clients, $T$ to denote the number of tasks in each client, $C$ to denote the number of classes in each task. We curate tasks by randomly selecting several classes from the datasets and sample part of the instances from these classes. Adhering to the principle of class incremental learning, there are no overlapped classes between any two tasks within a client.

**EMNIST-LTP.** The EMNIST dataset is a character classification dataset with 26 classes (Cohen et al., 2017). It contains 145600 instances of 26 English letters. The data contains both upper and lower case with the same label, making it more challenging for classification. To curate a dataset under LTP setting, we randomly sampled classes from the entire dataset for each client. The EMNIST-LTP dataset consists of 8 clients, with each client encompassing 6 tasks, each task comprising 2 classes ($N = 8, T = 6, C = 2$).

**EMNIST-shuffle.** In conventional reshuffling setting, the task sets are consistent across all clients, while arranged in different orders. Therefore, with the same structure as EMNIST-LTP, we construct EMNIST-shuffle dataset with 8 clients, 6 tasks each, and each task comprising 2 classes. While the 6 tasks of all clients are the same but in shuffled orders.

**EMNIST-noisy.** In this paper, we argue that forgetting is not invariably detrimental in FCL and propose the concept of accurate forgetting. To validate our argument and effectiveness of our method, we curate the EMNIST-noisy dataset with a few malicious clients by assigning random labels to the data. Besides, there could be noise in realistic dataset, including label noise. And malicious clients with adversarial behavior should also be taken into consideration under cross-device setting in Federate Learning. Robustness of FCL methods is crucial in real-world application. The EMNIST-noisy possesses the same structure as EMNIST-LTP dataset ($N = 8, T = 6, C = 2$). We randomly selects several clients and assign random labels to their first three tasks. After learning sequentially on all tasks, we evaluate on the last three tasks without noisy labels. By this means, we assess the impact of incorporating noisy information into the memory bank from previous tasks.

**CIFAR100.** As a challenging image classification dataset, CIFAR100 consists of low resolution images containing various objects and complex image backgrounds (Krizhevsky et al., 2009). We randomly sample 20 classes among 100 classes of CIFAR100 as a task for each of the 10 clients, and there are 4 tasks for each client ($N = 10, T = 4, C = 20$). For each class, we randomly sample 400 instances into the client dataset.

**MNIST-SVHN-F.** The mixed dataset is constructed with MNIST (LeCun et al., 1998), SVHN (Netzer et al., 2011) and FashionMNIST (Xiao et al., 2017). Similar to MNIST, SVHN dataset serves as a benchmark for digit classification tasks, notable for its representation of real-world scenarios with complex backgrounds. We unify the labels of these two datasets. FashionMNIST dataset is designed for clothing image classification. We set 10 clients in the mixed dataset, with each client containing 6 tasks, and each task has 3 classes. ($N = 10, T = 6, C = 3$). In this mixed dataset, different

tasks rely on different features. For example, shape features that are relevant to digit classification differ significantly from those that are important for classifying clothing items. If clients collaborate naively, it may result in a model that relies too heavily on spurious correlations, thus neglecting the significance of task-specific features.

## B  BASELINES

We compare our method `AF-FCL` with two baselines from FL, two baselines from CL and three baselines from FCL. The FL methods simply train a global model on sequential tasks, without any memorizing technique. The CL methods are respectively combined with the FL methods, training a global model while fighting catastrophic forgetting. The FCL methods focus on addressing the issues of catastrophic forgetting along with statistical heterogeneity.

**FedAvg** (McMahan et al., 2017). As a representative FL method, FedAvg trains the models in each client with local dataset and averages their parameters to attain a global model.

**FedProx** (Li et al., 2020). The algorithm is similiar to FedAvg. While training local models, a regularization term is employed to govern the proximity between the local parameters and the global parameters. This regularization term serves to effectively control the degree of deviation exhibited by the local models from the global model during the training process.

**PODNet** (Douillard et al., 2020). As a CL method, the algorithm incorporates a spatial-based distillation loss onto the feature maps of the classifier. This loss term serves to encourage the local models to align their respective feature maps with those of the previous model, thereby maintaining the performance in previous tasks.

**ACGAN-Replay**. This CL algorithm employs a GAN-based generative replay method (Wu et al., 2018). The algorithm trains an ACGAN in the data space to memorize the distribution of previous tasks. While learning on new tasks, the classifier is trained on new task data along with generated data from ACGAN.

**FLwF2T**. As a FCL algorithm, FLwF2T leverages the concept of knowledge distillation within the framework of federated learning (Usmanova et al., 2021). It employs both the old classifier from previous task and global classifier from server to train the local classifier.

**FedCIL**. The FCL algorithm extends the ACGAN-Replay method within the federated scenario, addressing the statistical heterogeneity issue with distillation loss (Qi et al., 2023).

**GLFC**. In FCL scenario, the algorithm exploits a distillation-based method to alleviate the issue of catastrophic forgetting from both local and global perspectives (Dong et al., 2022).

## C  IMPLEMENTATION DETAILS

### C.1  ALGORITHM

The algorithm of our method is detailed in Algorighm 1.

### C.2  METRICS

We use the metrics of accuracy and average forgetting for evaluation following recent works (Mirzadeh et al., 2021; Yoon et al., 2021a). Suppose $a_k^{t,i}$ is the test set accuracy of the $i-$th task after learning the $t-$th task in client $k$.

**Average Accuracy.** We evaluate the performance of the model on all tasks in all clients after it finish learning all tasks. By using a weighted average, we calculated the test set accuracy for all seen tasks across all clients, with the number of samples in each task serving as the weights:

$$\text{Average Accuracy} = \frac{1}{\sum_{k=1}^{N}\sum_{i=1}^{T} n_k^i} \sum_{k=1}^{N}\sum_{i=1}^{T} a_k^{T,i} * n_k^i. \tag{10}$$

This approach allows us to account for variations in task difficulty and ensure a fair evaluation across different tasks and clients.

---

**Algorithm 1** Federated continual learning framework `AF-FCL`

---

**Input:** Datasets of $T$ tasks for $N$ clients $\{D_1, D_2, \ldots, D_N\}$, $D_k = \{\mathcal{T}_k^1, \mathcal{T}_k^2, \ldots, \mathcal{T}_k^t\}$, classifier $h$ and normalizing flow model $g$;

1: **for** task $t = 1, 2, \ldots, T$ **do**
2:    $h' \leftarrow h$; $g' \leftarrow g$
3:    **for** round $r = 1, 2, \ldots$ **do**
4:       **Server** randomly selects clients $\mathcal{C}$ for local training and send them model parameters
5:       **for** client $\mathcal{C}_k \in \mathcal{C}$ **do**
6:          Optimize $g$ as in Eq. 5 with client dataset $\mathcal{D}_k^t$ and $g'$
7:          Calculate distribution parameters of client data with $g$ as in Eq. 7
8:          Generate features $\bar{u}_i$ with $g$ and perform likelihood estimation with above parameters
9:          Optimize $h$ as in Eq. 9 with client dataset, generated features, exact likelihood $p_{D_k^t}(\bar{u}_i)$ and $h'$
10:      **end for**
11:      the **Server** aggregates the parameters of $h_\theta^i$ and $g_\phi^i$ from clients $\mathcal{C}$ and weighted averages the parameters by client data number
12:    **end for**
13: **end for**
14: **Output:** the learned classification model $h$.

---

**Average Forgetting** The metric of average forgetting assesses the extend of backward transfer during continual learning, quantified as the disparity between the peak accuracy and the ending accuracy of each task. We also use a weighted average when calculating average forgetting:

$$\text{Average Forgetting} = \frac{1}{\sum_{k=1}^N \sum_{i=1}^{T-1} n_k^i} \sum_{k=1}^N \sum_{i=1}^{T-1} \max_{t \in \{1, \ldots, T-1\}} (a_k^{t,i} - a_k^{T,i}) * n_k^i. \quad (11)$$

## C.3   OPTIMIZATION

The Adam optimizer is employed for training all models. For all experiments except for CIFAR100, a learning rate of 1e-4 is utilized, with a global communication round of 60, and local iteration of 100. We set learning rate as 1e-3, global communication round as 40, and local iteration as 400 for CIFAR100. Consistent with prior research (Yoon et al., 2021a; Qi et al., 2023), all clients participate in each communication round. For training, a mini-batch size of 64 is adopted. The number of generated samples in an iteration aligns with this mini-batch size. We report the mean and standard deviation of each experiment, conducted three times with different random seed.

## C.4   MODEL ARCHITECTURES

In the case of CIFAR100, we utilize the feature extractor of a ResNet-18 (He et al., 2016) as $h_a$ and $h_b$ comprises two FC layers , both with 512 units. While for other datasets we adopt a three-layer CNN followed by a FC layer with 512 units as $h_a$. The channel numbers of the convolutional layers are $[64, 128, 256]$. And $h_b$ is represented by a FC layer. The outputs of $h_a$ belong to $\mathbb{R}^{512}$. All the FC layers employed in the architectures consist of 512 units. The convolutional layers and FC layers are followed by a Leaky ReLU layer. Another FC layer serves as $h_c$ and operates as the classification head.

The NF models consist of four layers of random permutation layer and affine coupling layer. The random permutation layers randomly permute the input vector so that various dependency among dimensions of input vectors could be effectively modeled. The inverse function of random permutation layers is to reversely permute the vector back to the original order. The affine coupling layers firstly partition the input vector into two halves $x_a$ and $x_b$. Then an affine transformation is applied to one part of the input, conditioned on the other part:

$$y_a = \exp(s(x_a)) \odot x_b + t(x_a), \quad (12)$$
$$y_b = x_b, \quad (13)$$

where $s$ and $t$ denote functions that create scaling and translation parameters, which we implemented with 2 blocks of residual neural network and learned from the data. The output vector $y$ is the concatenation of $y_a$ and $y_b$. The invertibility of affine coupling transformation is readily apparent.

# D  ADDITIONAL EXPERIMENTAL RESULTS

## D.1  ABLATION STUDIES

Our method consists of three major components: (I) feature generative-replay (GR). For generative replay, we train a global NF model in the feature space of classifier. By augmenting the learning process of classifier with the generated features, we prevent complete forgetting of previous tasks. (II) knowledge distillation (KD). The NF model is trained in the feature space of classifier. To maintain the stability of the training process for the NF model, a knowledge distillation loss is employed in the feature space of classifier, mitigating significant drift. (III) correlation estimation for accurate forgetting (AF). We assess the reliability of the generated feature by its probability density within the current tasks. Leveraging the NF model, we approximate the local feature distribution to evaluate the probability of a given generated feature aligning with the current distribution.

We conduct ablation studies on the EMNIST-LTP and EMNIST-shuffle dataset as displayed in Table 3. Our method achieves optimal performance with all the three modules. Without the GR module, the AF module also loses efficacy. Therefore, left with the KD module, the performance of our model is comparable to that of PODNet and FLwF2T which relies on knowledge distillation to retain previous knowledge. Without the AF module, our method degrades into naive generative reply based method, thus the performance is close to FedCIL and ACGAN-Replay.

Table 3: Ablation studies on EMNIST-LTP and EMNIST-shuffle dataset.

| Model | EMNIST-LTP | | EMNIST-shuffle | |
|---|---|---|---|---|
| | Accuracy↑ | Forgetting↓ | Accuracy↑ | Forgetting↓ |
| PODNet+FedAvg | $36.9_{\pm1.3}$ | $19.8_{\pm0.9}$ | $71.0_{\pm0.4}$ | $\mathbf{3.9}_{\pm0.4}$ |
| PODNet+FedProx | $40.4_{\pm0.4}$ | $14.3_{\pm0.5}$ | $70.6_{\pm0.7}$ | $9.6_{\pm0.3}$ |
| ACGAN-Replay+FedAvg | $38.4_{\pm0.2}$ | $9.8_{\pm0.8}$ | $70.0_{\pm0.5}$ | $4.7_{\pm0.3}$ |
| ACGAN-Replay+FedProx | $41.3_{\pm0.9}$ | $10.4_{\pm0.7}$ | $70.3_{\pm1.2}$ | $6.1_{\pm2.0}$ |
| FLwF2T | $40.1_{\pm0.3}$ | $15.5_{\pm0.5}$ | $71.0_{\pm0.9}$ | $8.1_{\pm0.8}$ |
| FedCIL | $42.0_{\pm0.6}$ | $12.4_{\pm0.3}$ | $71.1_{\pm0.4}$ | $6.4_{\pm0.2}$ |
| AF-FCL w/o GR | $38.8_{\pm1.5}$ | $15.3_{\pm0.4}$ | $70.8_{\pm0.7}$ | $6.7_{\pm0.5}$ |
| AF-FCL w/o KD | $44.3_{\pm0.6}$ | $10.7_{\pm0.7}$ | $72.1_{\pm0.5}$ | $5.8_{\pm0.3}$ |
| AF-FCL w/o AF | $41.8_{\pm0.3}$ | $13.7_{\pm1.2}$ | $71.0_{\pm0.9}$ | $6.7_{\pm0.4}$ |
| AF-FCL | $\mathbf{47.5}_{\pm0.3}$ | $\mathbf{7.9}_{\pm0.5}$ | $\mathbf{75.8}_{\pm0.2}$ | $4.2_{\pm0.1}$ |

## D.2  CIFAR100 IN A DIFFERENT SETTING

We conduct experiments on CIFAR100 with a more challenging setting. We randomly sample 10 classes among 100 classes of CIFAR100 as a task for each of the 8 clients, and there are 6 tasks for each client ($N = 8, T = 6, C = 10$). For each class, we randomly sample 400 instances into the client dataset. Therefore, each client possesses more tasks while less samples per task.

As shown in Table 4, our method attains the highest accuracy among the evaluated methods. Although the CL methods and conventional FCL methods emphasize the retention of knowledge acquired from previous tasks, indiscriminate memorization of potentially erroneous knowledge can detrimentally impact the performance on previous tasks. In contrast, our proposed method adopts a adaptive approach to forgetting biased features, resulting in a notable reduction of forgetting compared to established baselines, thus preserving a higher degree of task-specific knowledge retention.

## D.3  RESULTS OF EMNIST-NOISY DATASET

We conduct experiments in the EMNIST-noisy dataset with an increasing number of noisy clients. We display the complete comparison of accuracy and forgetting among baselines here. It is observed

Table 4: Average accuracy and forgetting on CIFAR100 when $N = 8, T = 6, C = 10$.

| Model | Accuracy↑ | Forgetting↓ |
|---|---|---|
| FedAvg | $19.5_{\pm0.3}$ | $2.4_{\pm0.20}$ |
| FedProx | $20.1_{\pm0.2}$ | $1.9_{\pm0.08}$ |
| PODNet+FedAvg | $21.3_{\pm0.1}$ | $2.0_{\pm0.06}$ |
| PODNet+FedProx | $21.6_{\pm0.4}$ | $2.1_{\pm0.15}$ |
| ACGAN-Replay+FedAvg | $19.5_{\pm0.6}$ | $3.0_{\pm0.36}$ |
| ACGAN-Replay+FedProx | $19.6_{\pm0.2}$ | $2.8_{\pm0.40}$ |
| FLwF2T | $21.5_{\pm0.7}$ | $5.9_{\pm0.67}$ |
| FedCIL | $19.6_{\pm0.3}$ | $2.9_{\pm0.52}$ |
| GLFC | $19.9_{\pm0.4}$ | $3.2_{\pm0.31}$ |
| AF-FCL | $\mathbf{23.8}_{\pm0.6}$ | $\mathbf{0.9}_{\pm0.07}$ |

that the performance of the methods consistently diminishes with the escalating count of noisy clients, as depicted in Table 5. The presence of noisy clients introduce harmful information into the model learning process and memorization of such information proves detrimental to the overall performance. Thus, some of the CL and FCL methods, which aim to fight forgetting, exhibit inferior performance compared to FL methods. Our approach employs adaptive mechanisms to mitigate the impact of erroneous information. By alleviating the negative influence of noisy clients, our method consistently surpasses all baselines in both accuracy and resistance to forgetting.

Table 5: Average accuracy and forgetting on EMNIST-noisy dataset in the last 3 tasks with different number of malicious clients $M$.

| Model | $M = 1$ | | $M = 2$ | | $M = 4$ | |
|---|---|---|---|---|---|---|
| | Accuracy↑ | Forgetting↓ | Accuracy↑ | Forgetting↓ | Accuracy↑ | Forgetting↓ |
| FedAvg | $52.3_{\pm0.7}$ | $16.1_{\pm0.9}$ | $51.7_{\pm0.5}$ | $16.0_{\pm0.4}$ | $50.4_{\pm0.6}$ | $12.1_{\pm0.8}$ |
| FedProx | $52.5_{\pm0.5}$ | $12.5_{\pm0.4}$ | $51.8_{\pm0.6}$ | $18.8_{\pm1.4}$ | $51.0_{\pm0.5}$ | $13.5_{\pm0.7}$ |
| PODNet+FedAvg | $43.3_{\pm1.3}$ | $20.3_{\pm0.7}$ | $38.5_{\pm0.9}$ | $20.1_{\pm0.2}$ | $33.8_{\pm0.7}$ | $19.0_{\pm0.9}$ |
| PODNet+FedProx | $44.3_{\pm0.6}$ | $19.6_{\pm0.7}$ | $37.3_{\pm1.3}$ | $21.2_{\pm0.8}$ | $34.1_{\pm1.3}$ | $18.4_{\pm0.6}$ |
| ACGAN-Replay+FedAvg | $45.8_{\pm0.6}$ | $18.6_{\pm0.5}$ | $42.6_{\pm0.9}$ | $17.5_{\pm0.6}$ | $40.2_{\pm0.9}$ | $16.0_{\pm0.9}$ |
| ACGAN-Replay+FedProx | $50.2_{\pm0.4}$ | $18.5_{\pm0.2}$ | $43.7_{\pm1.0}$ | $17.2_{\pm0.4}$ | $39.6_{\pm0.6}$ | $16.4_{\pm0.7}$ |
| FLwF2T | $52.1_{\pm0.7}$ | $14.7_{\pm2.3}$ | $47.6_{\pm0.3}$ | $18.6_{\pm1.9}$ | $44.5_{\pm0.5}$ | $14.1_{\pm0.3}$ |
| FedCIL | $49.8_{\pm0.4}$ | $15.2_{\pm0.9}$ | $45.8_{\pm0.7}$ | $19.1_{\pm0.5}$ | $42.0_{\pm0.8}$ | $15.8_{\pm1.4}$ |
| AF-FCL | $\mathbf{55.5}_{\pm0.5}$ | $\mathbf{7.5}_{\pm0.8}$ | $\mathbf{54.9}_{\pm0.4}$ | $\mathbf{11.8}_{\pm0.5}$ | $\mathbf{54.0}_{\pm0.6}$ | $\mathbf{12.8}_{\pm0.7}$ |

## D.4 RESULTS OF IMAGENET-SUBSET DATASET

We conducted experiments on a subset of the ImageNet dataset. Each client among 10 clients contains 4 tasks, where each task consists of 40 classes among 200 classes. As shown in the table below, our method surpasses existing baselines. This empirical evidence demonstrates the efficacy of our method, particularly in handling richer semantic information on large datasets such as ImageNet.

Table 6: Average accuracy and forgetting on ImageNet-Subset dataset when $N = 10, T = 4, C = 40$.

| Model | Accuracy↑ | Forgetting↓ |
|---|---|---|
| FedAvg | 14.7 | 3.2 |
| FedProx | 15.1 | 2.3 |
| ACGAN-Replay+FedAvg | 17.4 | 1.6 |
| ACGAN-Replay+FedProx | 17.3 | 1.8 |
| FedCIL | 17.8 | **1.2** |
| GLFC | 18.0 | 1.9 |
| AF-FCL | **20.4** | 1.7 |

# E    COMPUTATION ANALYSIS AND DEVICES

As a generative-replay based model, `AF-FCL` has a similar number of parameters with other generative-replay based methods, including the baselines FedCIL, ACGAN-Replay, etc. Due to the special design of NF models, the generation and density estimation of them are fast and efficient. Therefore, `AF-FCL` does not bring many extra computational and communication costs. We provide the running-time comparisons with baselines in Table 7. As shown in the table, running-time of the proposed method is less than that of the generative-replay based models mentioned above.

**Devices** In the experiments, we conduct all methods on a local Linux server that has two physical CPU chips (Intel(R) Xeon(R) CPU E5-2640 v4 @ 2.40GHz) and 32 logical kernels. All methods are implemented using Pytorch framework and all models are trained on GeForce RTX 2080 Ti GPUs.

Table 7: Run-time consumption comparisons on the EMNIST-LTP and CIFAR100 dataset

| Methods | Run-time consumption (EMNIST-LTP) | Run-time consumption (CIFAR100) |
|---|---|---|
| FedAvg | 22 min | 238 min |
| FedProx | 26 min | 245 min |
| PODNet+FedAvg | 35 min | 252 min |
| PODNet+FedProx | 37 min | 253 min |
| ACGAN-Replay+FedAvg | 85 min | 312 min |
| ACGAN-Replay+FedProx | 89 min | 315 min |
| FLwF2T | 33 min | 248 min |
| FedCIL | 93 min | 322 min |
| AF-FCL | 62 min | 302 min |

# F    RELATED NOTIONS

## F.1    BIASED FEATURES

Researchers have employed various definitions for biased features, one of which involves defining them as spurious correlations. We denote $\mathcal{X}, \mathcal{Y}$ as an input and output space of machine learning algorithm. An algorithm learns a mapping from the data $x \in \mathcal{X}$ to the prediction $\hat{y} \in \mathcal{Y}$: $\hat{y} = f(x)$. We assume there are attributes $\gamma_1, \gamma_2, ...$ abstracted from the data $x$. For example, $\gamma_1$ represents the shape of the object in the input image $x$, and $\gamma_2$ denotes the number of black pixels in the input image $x$. The machine learning algorithm actually relies on many attributes to conduct infering: $\hat{y} = f(\gamma_{i_1}, \gamma_{i_2}, ..., \gamma_{i_N})$. We define an attribute $\gamma$ as biased feature if it does not comply with the natural meaning of the target $y$ Jeon et al. (2022). Relying on such biased attribute would result in poor generalizability of the algorithm. The biased features could be attained through biased training dataset and the learned mapping $f$ relying on the biased features may not perform well in the testing dataset. For instance, if in the training image dataset all cows are standing on the grass, the machine learning model may rely on the attribute 'grass' for classifying images of cows.

In Sec. 4, we instantiate biased features with label noise (Zhang et al., 2021; Chen et al., 2020). With random labels, the model probably extracts misaligned attributes. In benchmark datasets, machine learning models may also learn biased features even without label noise (Zhu et al., 2021a).

## F.2    CONCEPT DRIFTS

Different from the studies about Federated Continual Learning, the evaluation in the concept drift studies is conducted at each time step. Therefore, there is no memorization requirement or catastrophic forgetting problem in the concept drift studies. A novel clustering algorithms for reacting to concept drifts is proposed (Jothimurugesan et al., 2023). Adaptive-FedAVG adapted the learning rate to react to concept drift (Canonaco et al., 2021). Panchal et al. proposed to detect concept drift through the magnitude of parameter updates and designed a novel adaptive optimizer Panchal et al. (2023).

### F.3    ORTHOGONAL TRAINING

The incorporation of orthogonal training and our accurate forgetting method is a promising direction. Bakman et al. proposed to modify the subspace of model layers in learning new tasks such that it is orthogonal to the global principal subspace of old tasks (Bakman et al., 2023). By distinguishing the subspace inside the model for each task, catastrophic forgetting of old tasks is mitigated, and it also relieves the influence of unrelated tasks. We will continue to explore the employment of orthogonal training in our method.

Our method explicitly quantifies the correlations of generated features through probability calculations. Moreover, we facilitate selective forgetting by assigning lower weights to erroneous old knowledge, thus enabling the classifier to discard biased features and achieve improved overall performance.

