# OpenReview forum: "Accurate Forgetting for Heterogeneous Federated Continual Learning"
_ICLR.cc/2024/Conference — ICLR 2024 poster_

### Official Review · Reviewer_hHgp · 2023-10-31

**Soundness:** 2 fair
**Presentation:** 3 good
**Contribution:** 2 fair
**Rating:** 6
**Confidence:** 4

**Summary:**

The paper presents an interesting and novel approach to addressing the challenges of federated continual learning (FCL), particularly in scenarios where data and tasks among different clients are potentially unrelated or even antagonistic. The concept of "accurate forgetting" (AF) and the proposed AF-FCL method are well-motivated and empirically evaluated. The paper is well-written and addresses an important problem in the intersection of federated learning and continual learning.

**Strengths:**

1. The paper introduces the novel concept of "accurate forgetting" (AF) in the context of FCL. It is commendable to challenge the conventional wisdom that forgetting is invariably detrimental and instead show that in specific situations, it can be beneficial.

2. The paper conducts comprehensive experiments to evaluate the proposed method against various baselines. The results show the superiority of the AF-FCL method, which strengthens the paper's contributions.

3. The paper is well-structured and clearly written.

**Weaknesses:**

1. The authors' use of feature correlation for replay training is intriguing, though the batch-wise weighting method employed seems somewhat simplistic. Recent research has explored orthogonal training techniques [1] to mitigate the influence of uncorrelated or biased old features. Consequently, a more nuanced feature weighting mechanism might yield improved results.

2. It would be beneficial for the authors to provide a more detailed rationale for the application of the Normalization Flow (NF) model in this context. While any generative model could potentially be used to generate old features with a suitable feature extractor, it is unclear why the NF model was chosen specifically. It may be valuable to conduct experiments to demonstrate the necessity of the NF model in comparison to other generative models.

3. Additional information regarding the architecture of the NF model and the communication cost it incurs would enhance the clarity and practicality of the proposed method.

4. To better simulate the non-iid setting in Federated Learning, it is suggested that the authors consider employing a Dirichlet distribution [2], which could be more representative of real-world data distribution patterns than the current setting.

5. While the paper addresses "challenging" datasets, such as CIFAR100 and MNIST-SVHN-F, it is recommended that the authors conduct experiments on truly challenging datasets like ImageNet. Additionally, the paper should include experiments with shuffled task orders for CIFAR100 and other challenging datasets to provide a more comprehensive evaluation of the proposed method.

6. It is advisable for the authors to include comparisons with more recent baseline methods, such as TARGET (ICCV 2023 [3]), to ensure that the proposed approach is benchmarked against the latest state-of-the-art techniques in the field.

[1] Bakman, Yavuz Faruk, et al. "Federated Orthogonal Training: Mitigating Global Catastrophic Forgetting in Continual Federated Learning." arXiv preprint arXiv:2309.01289 (2023).

[2] Hsu, Tzu-Ming Harry, Hang Qi, and Matthew Brown. "Measuring the effects of non-identical data distribution for federated visual classification." arXiv preprint arXiv:1909.06335 (2019).

[3] Zhang, Jie, et al. "TARGET: Federated Class-Continual Learning via Exemplar-Free Distillation." Proceedings of the IEEE/CVF International Conference on Computer Vision. 2023.

**Questions:**

Please see the weaknesses

---

> ### Author Response · Authors · 2023-11-17
> **Response to Reviewer hHgp (part 1)**
>
> We would like to thank the reviewer for the insightful comments. Below are our responses to the comments in **Weaknesses** and **Questions**.
>
> ---
>
> #### ***Question 1: The authors' use of feature correlation for replay training is intriguing, though the batch-wise weighting method employed seems somewhat simplistic. Recent research has explored orthogonal training techniques to mitigate the influence of uncorrelated or biased old features. Consequently, a more nuanced feature weighting mechanism might yield improved results.***
> ****Answer:**** Thanks for the insightful advice. The incorporation of orthogonal training and our accurate forgetting method is a promising direction. We have supplemented the discussion in the paper and cited the work of orthogonal training in the FCL scenario [1].
>
> [1] proposed to modify the subspace of model layers in learning new tasks such that it is orthogonal to the global principal subspace of old tasks. By distinguishing the subspace inside the model for each task, catastrophic forgetting of old tasks is mitigated, and it also relieves the influence of unrelated tasks. We will continue to explore the employment of orthogonal training in our method.
>
> Our method explicitly quantifies the correlations of generated features through probability calculations. Moreover, we facilitate selective forgetting by assigning lower weights to erroneous old knowledge, thus enabling the classifier to discard biased features and achieve improved overall performance. We are happy to know that orthogonal training techniques could mitigate the influence of uncorrelated or biased old features. We will continue this topic in the future.
>
> [1] Bakman, Yavuz Faruk, et al. "Federated Orthogonal Training: Mitigating Global Catastrophic Forgetting in Continual Federated Learning." arXiv preprint arXiv:2309.01289 (2023).
>
> ---
>
> #### ***Question 2: It would be beneficial for the authors to provide a more detailed rationale for the application of the Normalization Flow (NF) model in this context. It is unclear why the NF model was chosen specifically. It may be valuable to conduct experiments to demonstrate the necessity of the NF model in comparison to other generative models.***
> ****Answer:****
>
> We would like to thank the reviewer for the professional comments. We would like to explain it as follows:
>
> 1. **The NF model is able to accurately estimate the probability density of observed data.** With such unique capabilities absent in other generative models, we evaluate the benefits of generating features for the current task and achieve accurate forgetting through probabilistic estimation.
>
> 2. **NF model could map an arbitrarily complex data distribution to a pre-defined distribution losslessly through a sequence of bijective transformations.** Such invertability enables the NF to have a lossless memory of the input knowledge.
>
> 3. **We have supplemented ablation studies validating that the NF model is superior to the GAN model in our method**; thereby, the correlation estimation module of our method also loses effects. As the results shown in the table below, AF-FCL-GAN degrades into naive generative-reply based method, thus the performance has significantly declined.
>
> | Model | EMNIST-LTP | EMNIST-LTP | CIFAR100 | CIFAR100 |
> | --- |  --- |  --- |  --- |  --- |
> |  | Accuracy | Forgetting | Accuracy | Forgetting |
> | AF-FCL (GAN) | 41.5 | 12.2 | 32.4 | 7.5 |
> | AF-FCL (NF) | **47.5** | **7.9** | **36.3** | **4.9** |
>
> ---

---

> > ### Author Response · Authors · 2023-11-17
> > **Response to Reviewer hHgp (part 2)**
> >
> > #### ***Question 3: Additional information regarding the architecture of the NF model and the communication cost it incurs would enhance the clarity and practicality of the proposed method.***
> > ****Answer:**** Thanks for the valuable advice. We have supplemented the details in the paper.
> >
> > 1. **The NF models consist of four layers of random permutation layers and affine coupling layers.** The random permutation layers randomly permute the input vector so that various dependency among dimensions of input vectors could be effectively modeled. The inverse function of random permutation layers is to reversely permute the vector back to its original order. The affine coupling layers first partition the input vector into two halves, $x_a$ and $x_b$. Then an affine transformation is applied to one part of the input, conditioned on the other part:
> > $$
> > \begin{split}
> > y_a&=\exp(s(x_a))\odot x_b+t(x_a),\\
> > y_b&=x_b,
> > \end{split}
> > $$
> > where $s$ and $t$ denote functions that create scaling and translation parameters, which we implemented with 2 blocks of residual neural network and learned from the data. The output vector $y$ is the concatenation of $y_a$ and $y_b$. The invertibility of affine coupling transformations is readily apparent.
> >
> > 2. **The communication mode of AF-FCL is the same as FedAvg.** After training locally, the clients send gradients of the model to the server. Then the server transmits the aggregated gradients back to clients. Therefore, same as other baselines, the communication cost for AF-FCL is the number of clients $\times$ number of communication rounds $\times$ number of model parameters. The number of parameters in NF models from AF-FCL and in GAN models from FedCIL is displayed in the table below. The number of parameters in NF models is comparable to that in GAN models.
> >
> > | | EMNIST | CIFAR100 |
> > | --- | --- | --- |
> > | FedCIL | 4.993 M | 5.902 M |
> > | AF-FCL | 5.943 M | 6.398 M |
> >
> > ---
> >
> > #### ***Question 4: To better simulate the non-iid setting in Federated Learning, it is suggested that the authors consider employing a Dirichlet distribution, which could be more representative of real-world data distribution patterns than the current setting.***
> > ****Answer:**** Thanks for the insightful and practical comments. Following the advice, we supplement experiments on the EMNIST dataset distributed with a Dirichlet distribution. We partition the dataset into 48 tasks with Dirichlet distributions $Dir_{48}(0.1)$ and $Dir_{48}(0.1)$ following the work in [2]. And we randomly assign 6 tasks to each of the 8 clients. The results of our method and baselines are shown in the table below. Under a less heterogeneous setting ($Dir_{48}(0.5)$), clients collaborate more closely with each other, thus the accuracy is higher. AF-FCL outperforms all baselines under the Dirichlet distribution setting.
> >
> > | Model | $Dir_{48}(0.1)$ | $Dir_{48}(0.1)$ | $Dir_{48}(0.5)$ | $Dir_{48}(0.5)$ |
> > | --- |  --- |  --- |  --- |  --- |
> > |  | Accuracy | Forgetting | Accuracy | Forgetting |
> > | FedAvg | 47.3 | 13.4 | 63.9 | 8.7 |
> > | FedProx | 47.6  | 13.7 | 63.8 | 8.2 |
> > | PODNet+FedAvg | 51.0 | 10.7 | 63.9 | 8.3 |
> > | PODNet+FedProx | 50.7 | 10.6 | 64.6 | 7.5 |
> > | ACGAN-Replay+FedAvg | 52.4 | 9.1 | 66.7 | 5.2 |
> > | ACGAN-Replay+FedProx | 54.7 | 8.3 | 66.0 | 5.9 |
> > | FLwF2T | 50.2 | 11.8 | 66.3 | 5.0 |
> > | FedCIL | 55.0 | 8.7 | 66.8 | 4.9 |
> > | GLFC | 52.2 | 9.3 | 65.3 | 5.1 |
> > | AF-FCL | **59.7** | **8.0** | **70.0** | 5.1 |
> >
> > [2] Wang, Jianyu, et al. Tackling the objective inconsistency problem in heterogeneous federated optimization. Advances in neural information processing systems 33 (2020): 7611-7623.
> >
> > ---

---

> > > ### Author Response · Authors · 2023-11-17
> > > **Response to Reviewer hHgp (part 3)**
> > >
> > > #### ***Question 5: While the paper addresses "challenging" datasets, such as CIFAR100 and MNIST-SVHN-F, it is recommended that the authors conduct experiments on truly challenging datasets like ImageNet. Additionally, the paper should include experiments with shuffled task orders for CIFAR100 and other challenging datasets to provide a more comprehensive evaluation of the proposed method.***
> > >
> > > ****Answer:**** Thanks for the valuable advice. We would like to explain as follows:
> > >
> > > 1. By "challenging datasets," we refer to those relative to existing work. **We consider a practical setting LTP where two clients may not share any common tasks.** We have also expanded our experiments to include other challenging datasets following the advice, as described below.
> > >
> > > 2. **Following your suggestions, we have supplemented experiments on CIFAR100 dataset with LTP setting: CIFAR100-LTP.** By randomly sampling 10 classes as a task among the 100 classes of CIFAR100, we construct 6 tasks for each of the 8 clients. The results in the Table below demonstrate that our method achieves superior accuracy relative to baselines. While CL methods and traditional FCL approaches focus on preserving previously acquired knowledge, they often inadvertently retain inaccurate information, adversely affecting performance on preceding tasks. Conversely, our method implements an adaptive strategy for selectively forgetting biased features. Therefore, in such a challenging setting with high statistical heterogeneity, our approach significantly diminishes forgetting, outperforming existing baselines, and thereby enhancing the retention of task-specific knowledge.
> > >
> > > 3. We curate the CIFAR100-shuffle dataset and supplement experiments on it. As in the CIFAR100 dataset, we sample 20 classes among 100 classes as a task for each of the 10 clients, and there are 4 tasks for each client, but the task sets are consistent across all clients while arranged in different orders. The accuracy of the methods is displayed in the table below. The CIFAR100-shuffle dataset offers a more feasible option within the conventional setting, yielding higher accuracy rates, as shown in the accompanying table. Moreover, within this frequently adopted dataset setting, our method consistently exhibits superior performance compared to all baseline approaches.
> > >
> > >
> > > | Model | CIFAR100 | CIFAR100-LTP | CIFAR100-shuffle |
> > > | --- | --- | --- | --- |
> > > | FedAvg | 26.3 | 19.5 | 48.0 |
> > > | FedProx | 28.7 | 20.1 | 44.8 |
> > > | PODNet+FedAvg | 30.5 | 21.3 | 48.5 |
> > > | PODNet+FedProx | 32.5 | 21.6 | 42.5 |
> > > | ACGAN-Replay+FedAvg | 32.1 | 19.5 | 49.1 |
> > > | ACGAN-Replay+FedProx | 31.8 | 19.6 | 49.0 |
> > > | FLwF2T | 30.2 | 21.5 | 48.7 |
> > > | FedCIL | 33.5 | 19.6 | 49.3 |
> > > | GLFC | 35.6 | 19.9 | 48.2 |
> > > | AF-FCL | **36.3** | **23.8** | **51.3** |
> > >
> > > ---
> > >
> > > #### ***Question 6: It is advisable for the authors to include comparisons with more recent baseline methods, such as TARGET, to ensure that the proposed approach is benchmarked against the latest state-of-the-art techniques in the field.***
> > > ****Answer:**** Thanks for the valuable advice. TARGET is a generative-replay based method designed for federated continual learning. A global generator is trained to produce synthetic data to mitigate catastrophic forgetting of previous tasks [3]. We add TARGET as an extra baseline and supplement experiments on the EMNIST-LTP, EMNIST-shuffle, CIFAR100, and MNIST-SVHN-F datasets. As the results shown in the tables below, TARGET achieves comparable performance as another generative-replay based baseline FedCIL. Our method selectively utilizes previous knowledge and consistently outperforms other baselines.
> > >
> > > | Model | EMNIST-LTP | EMNIST-LTP | EMNIST-shuffle | EMNIST-shuffle |
> > > | --- | --- | --- | --- |  --- |
> > > | | Accuracy | Forgetting | Accuracy | Forgetting |
> > > | FLwF2T |  40.1 | 15.5 | 71.0 | 8.1 |
> > > | FedCIL | 42.0 | 12.4 | 71.1 | 6.4 |
> > > | GLFC |  40.1 | 14.3 | 74.9 | 5.6 |
> > > | **TARGET** | 40.9 | 13.6 | 71.2 | 6.0 |
> > > | AF-FCL | **47.5** | **7.9** | **75.8** | **4.2** |
> > >
> > > | Model | CIFAR100 | CIFAR100 |  MNIST-SVHN-F |  MNIST-SVHN-F |
> > > | --- | --- | --- | --- |  --- |
> > > | | Accuracy | Forgetting | Accuracy | Forgetting |
> > > | FLwF2T |  30.2 | 7.2 | 54.2 | 25.6 |
> > > | FedCIL | 33.5 | 6.5 | 57.2 | 19.7 |
> > > | GLFC |  35.6 | 6.2 | 61.8 | 10.8 |
> > > | **TARGET** | 32.2 | 6.4 | 55.4 | 20.7 |
> > > | AF-FCL | **36.3** | **4.9** | **68.1** | **7.5** |
> > >
> > > [3] Zhang, Jie, et al. TARGET: Federated Class-Continual Learning via Exemplar-Free Distillation. Proceedings of the IEEE/CVF International Conference on Computer Vision. 2023.

---

> ### Author Response · Authors · 2023-11-19
> **Need further Clarification?**
>
> Dear reviewer hHgp:
>
> We would like to appreciate your precious efforts in reviewing our paper. Meanwhile, we have done our best to clarify all your concerns in our rebuttal. For example, for the issue you are concerned about: **rationale for the application of the Normalization Flow (NF) model**, we would like to clarify that we utilize the unique characteristic of probability estimation of the NF model in our method, which we elaborate on in the answers to **Question 2**.
>
> Therefore, would you mind checking our response, and is there any unclear point so that we could further clarify?
>
> Best regards,
>
> Authors

---

> > ### Comment · Reviewer_hHgp · 2023-11-20
> >
> > Thanks for the detailed response. Although the response has clarified some of my concerns, I still hesitate to be positive due to the following issues.
> >
> > 1. As the communication cost you show, the NF model does introduce additional costs even compared to the GAN. Although leveraging the NF model seems to bring some performance gain, I don't think the additional communication cost is worth it. Besides, I noticed that the NF model parameters are distinct for different datasets, does that mean different datasets use different NF model architectures? Furthermore, using and communicating the generative model in FL still have a common problem, which is that the private information embedded in the generative model may divulge.
> >
> > 2. My suggestion of testing on larger and more challenging datasets does not question your task challenges. Experiments on larger datasets, like ImageNet, at least ImageNet-Subset, can test whether the NF model can handle and model more complicated data distribution, and can also test the effectiveness of your AF model when dealing with more informative semantics within the representation space.
> >
> > 3. I still think orthogonal learning is somewhat more elegant than the AF model since if orthogonal learning can be achieved, there is no need to mitigate the biased or harmful knowledge transfer from learned tasks to current ones. Besides, orthogonal learning does not need a generative model for replay learning.

---

> > > ### Author Response · Authors · 2023-11-21
> > > **Response to further questions of Reviewer hHgp**
> > >
> > > #### ***1.1: The NF model does introduce additional costs even compared to the GAN***
> > > **We would like to clarify that although the NF model has relatively more parameters in the response to Question 3, our method indeed has fewer communication costs compared with the GAN.**
> > > In spite of the NF model having marginally more parameters than the GAN model (see the answer to **Question3**), the slower convergence of GANs necessitates a higher number of communication rounds. Consequently, the communication cost of our method is lower than that of alternative approaches. For instance, the total communication cost of the baseline method FedCIL is 599.2 M in the EMNIST-LTP dataset, while the communication cost of our method is 534.8 M.
> > >
> > > #### ***1.2: I noticed that the NF model parameters are distinct for different datasets, does that mean different datasets use different NF model architectures?***
> > >
> > > **In our study, we utilized fundamentally identical NF models across various datasets.** Following existing works [1], the generative model is conditioned on the classes of the input samples. The slight variation in the number of parameters is attributed to the differing number of classes in each dataset, resulting in distinct conditional inputs for the NF models. All NF models consist of four layers of random permutation layers and affine coupling layers.
> > >
> > > #### ***1.3: Furthermore, using and communicating the generative model in FL still have a common problem, which is that the private information embedded in the generative model may divulge.***
> > >
> > > **We argee with the point that the generative replay methods in FCL may have a common problem.** Nevertheless, generative replay methods are more practical and effective. This is why it has received the attention of many researchers [1,2,3]. For example, [1] proposed learning a global generative model to mitigate catastrophic forgetting in FCL. Compared with the related baselines, we incorporate more considerations regarding privacy. By concealing the local feature extrator, which is the only model processing raw data, we can better protect client privacy for generative-replay based methods. In the future, we will continue to explore this issue for better privacy protection.
> > >
> > > #### ***2: Supplyment experiments on larger datasets, like ImageNet, at least ImageNet-Subset.***
> > >
> > > Following the kind advice, we conducted experiments on a subset of the ImageNet dataset. Each client among 10 clients contains 4 tasks, where each task consists of 40 classes among 200 classes.
> > > As shown in the table below, our method surpasses existing baselines. This empirical evidence demonstrates the efficacy of our method, particularly in handling richer semantic information on large datasets such as ImageNet.
> > >
> > > | Model | Accuracy | Forgetting |
> > > | --- |  --- |  --- |
> > > | FedAvg | 14.7 | 3.2 |
> > > | FedProx | 15.1 | 2.3 |
> > > | ACGAN-Replay+FedAvg | 17.4 | 1.6 |
> > > | ACGAN-Replay+FedProx | 17.3 | 1.8 |
> > > | FedCIL | 17.8 | 1.2 |
> > > | GLFC | 18.0 | 1.9 |
> > > | AF-FCL | 20.4 | 1.7 |
> > >
> > > #### ***3: I still think orthogonal learning is somewhat more elegant than the AF model since if orthogonal learning can be achieved, there is no need to mitigate the biased or harmful knowledge transfer from learned tasks to current ones.***
> > >
> > > **We would like to explain that, though orthogonal learning is a promising research direction, it indeed possesses inherent drawbacks compared with generative replay.** As you mentioned, if orthogonal learning **can be achieved**, it can obviate the need for generative models, thereby offering enhanced privacy protection. However, existing work indicates that orthogonal methods pose implementation challenges in practical applications [4]. In contrast, generative approaches are more straightforward to implement and yield superior results, which explains their widespread adoption in CL and FCL [1,2,3]. Besides, orthogonal methods often require more network parameters to store orthogonal representations, which increases when the number of tasks increases, making it hard when there are a large number of tasks [5].
> > >
> > > [1] Daiqing Qi, Handong Zhao, and Sheng Li. Better generative replay for continual federated learning. In The Eleventh International Conference on Learning Representations, ICLR 2023, Kigali, Rwanda, May 1-5, 2023. OpenReview.net, 2023.
> > > [2] Zhang, Jie, et al. TARGET: Federated Class-Continual Learning via Exemplar-Free Distillation. Proceedings of the IEEE/CVF International Conference on Computer Vision. 2023.
> > > [3] Shin, Hanul, et al. Continual learning with deep generative replay. Advances in neural information processing systems 30 (2017).
> > > [4] De Lange, Matthias, et al. A continual learning survey: Defying forgetting in classification tasks. IEEE transactions on pattern analysis and machine intelligence 44.7 (2021): 3366-3385.
> > > [5] Wang, Xiao, et al. Orthogonal Subspace Learning for Language Model Continual Learning. arXiv preprint arXiv:2310.14152 (2023).

---

> > > > ### Comment · Reviewer_hHgp · 2023-11-21
> > > >
> > > > Thanks for your quick reply. Although I cannot agree with all your ideas, I would like to increase my rating to 6. But do remember to include the experiment results on ImageNet or its subset with shuffling task order and standard task order (associated with existing FCL works) in the later version (revision or camera-ready), because they are very important.

---

> > > > > ### Author Response · Authors · 2023-11-22
> > > > > **Many thanks for raising the score**
> > > > >
> > > > > Thank you very much for all the suggestions you have provided. They are both deep and enlightening, playing a crucial role in refining and enhancing the quality of our paper. Experiments on ImageNet with different task settings are important for validating the effectiveness of our method when dealing with more informative semantics and will be included in the final version. Suggestions about orthogonal training are very insightful and enlightening. We will revise our paper in the final version according to all the comments provided.

---

### Official Review · Reviewer_T35E · 2023-10-31

**Soundness:** 3 good
**Presentation:** 3 good
**Contribution:** 3 good
**Rating:** 8
**Confidence:** 4

**Summary:**

In this paper, the authors discuss the problem of federated continual learning (FCL). Their solution forgetting problem in FCL is to train an NF Model (a type of generative model). This generative model has two benefits: first, it helps the clients to calculate the distribution parameters of the client data; second, it generates features that will be exploited to train the global model. The authors also propose a new task transition called limitless task pool (LTP), which is more suitable for federated settings. In LTP, clients' tasks are independently selected, and they may or may not share any data.

**Strengths:**

* The paper is well-motivated.
* The solution is novel but complex.
* The paper considers realistic scenarios, and it can outperform the prior works.
* Paper includes various datasets, prior works, comprehensive ablation on the different components of the loss function, and computation analysis.

**Weaknesses:**

* It is not straightforward to understand how difficult the baselines are because FedAvg and FedProx, without any continual learning mitigation mechanism, have about 8% forgetting. Maybe this is due to the fact that the performance of the models is not good even in the current tasks.
* In all the experiments, N (number of clients) is very small (only 10).
* Please also check out the questions.

**Questions:**

1- Do task transitions for the clients happen simultaneously, or do tasks change in different clients independently?

2- Is any example saved in the memory for AF-FCL? What is the memory size for the memory-based baselines (GFCL and FLwF2T)?

3- How robust is AF-FCL to data heterogeneity? Could you please include the information regarding how heterogeneous clients' data distribution is?

4- What is the communication cost for AF-FCL?

5- What is the number of parameters in NF models?

6- Are all the clients participating in the training every round?

7- How is g initialized? What is the forgetting in the NF model?

---

> ### Author Response · Authors · 2023-11-17
> **Response to Reviewer T35E (part 1)**
>
> We would like to thank the reviewer for the insightful comments. Below are our responses to the comments in **Weaknesses** and **Questions**.
>
> ---
>
> #### ***Question 1: It is not straightforward to understand how difficult the baselines are because FedAvg and FedProx, without any continual learning mitigation mechanism, have about 8% forgetting. Maybe this is due to the fact that the performance of the models is not good even in the current tasks.***
> ****Answer:**** Thanks for the valuable comments. We would like to explain as follows:
>
> 1. **We fine-tuned different hyper-parameters to guarantee convergency of each task and achieved optimal performance.** As shown in the table below, we chose a local iteration count of 400 as it leads to near convergence in our experiments.
>
> | local iteration | accuracy of task 1 |
> | --- | --- |
> | 100 | 13.7 |
> | 200 | 19.5 |
> | 400 | 22.9 |
> | 600 | 22.8 |
>
> 2. **The relatively low degree of forgetting observed can be attributed to the complexity of each task, combined with the collaboration among clients, which mitigates the forgetting effect.** The CIFAR100 dataset is a challenging dataset for the FCL problem. In this context, the federated model is tasked with classifying 100 distinct classes, where the baseline accuracy of random guessing stands at 1%. Consequently, achieving 26.3% accuracy with the FedAvg baseline represents a reasonable performance benchmark. Models trained on limited data for a single task perform poorly; however, when they learn more tasks and collaborate with other users, their performance relatively improves, mitigating the impact of catastrophic forgetting. For example, as the accuracy of each task in each time step shown in the table below, the accuracy of task 1 increases after the model is trained on the second task.
>
> | FedAvg | step 1 | step 2 | step 3 | step 4 |
> | --- | --- | --- | --- | --- |
> | task 1 | 22.9 | 26.8 | 22.7 | 18.9 |
> | task 2 | - | 25.9 | 30.0 | 20.0 |
> | task 3 | - | - | 35.5 | 27.7 |
> | task 4 | - | - | - | 38.5 |
>
> | AF-FCL | step 1 | step 2 | step 3 | step 4 |
> | --- | --- | --- | --- | --- |
> | task 1 | 20.34 | 24.1 | 30.4 | 29.5 |
> | task 2 | - | 29.2 | 34.7 | 29.9 |
> | task 3 | - | - | 45.1 | 36.1 |
> | task 4 | - | - | - | 49.9 |
>
> ---
>
> #### ***Question 2: In all the experiments, N (number of clients) is very small (only 10).***
> ****Answer:**** Thanks for the practical comments. We would like to explain as follows:
>
> 1. We follow existing work in FCL for consistency. The number of clients is set as 5 in related works [1].
>
> 2. **The dataset with a small number of clients is more challenging in the FCL scenario.** When there are numerous clients, a client is highly likely to spot collaborative clients with the same current task and previous tasks, which relieves the forgetting problem in FCL.
>
> 3. **We supplement experiments on the EMNIST-LTP dataset with more clients** ($N$ denotes the number of clients) as displayed in the table below. As previously noted, an increase in the number of clients results in diminished forgetting across all methods, thereby enhancing accuracy. Moreover, naive FL baselines, such as FedAvg and FedProx, exhibit performance comparable to those with explicit memorization techniques when the client count surpasses 14. Our method, which more effectively manages statistical heterogeneity among clients, consistently outperforms these baselines in most cases.
>
> | Model | $N=8$ | $N=8$ | $N=14$ | $N=14$ | $N=20$ | $N=20$ | $N=40$ | $N=40$ |
> | --- |  --- |  --- | --- |  --- | --- |  --- | --- |  --- |
> | | Accuracy | Forgetting | Accuracy | Forgetting | Accuracy | Forgetting | Accuracy | Forgetting |
> | FedAvg | 32.5 | 20.8 | 52.3 | 4.1 | 57.2 | 4.1 | 67.2 | 2.1 |
> | FedProx | 35.3 | 19.2 | 53.6 | 3.8 | 55.0 | 3.3 | 68.1 | 2.0
> | PODNet+FedAvg | 36.9 | 19.8 | 51.8 | 5.0 | 52.6 | 3.7 | 67.5 | 2.4
> | PODNet+FedProx | 40.4 | 14.3 | 54.0 | 4.6 | 59.9 | **3.2** | 66.5 | 2.8 |
> | ACGAN-Replay+FedAvg | 38.4 | 9.8 | 52.6 | 4.9 | 55.6 | 3.6 | 64.2 | 3.1 |
> | ACGAN-Replay+FedProx | 41.3 | 10.4 | 54.4 | 6.5 | 57.7 | 3.4 | 68.1 | 4.7 |
> | FLwF2T | 40.1 | 15.5 | 51.1 | 4.5 | 57.1 | 4.9 | 70.5 | 2.0 |
> | FedCIL | 42.0 | 12.4 | 55.0 | 4.9 | 56.2 | 4.2 | 68.0 | 3.2 |
> | GLFC | 40.1 | 14.3 | 51.3 | 5.0 | 57.3 | 4.6 | 69.3 | 1.6 |
> | AF-FCL | **47.5** | **7.9** | **64.6** | **3.5** | **67.1** | 3.9 | **73.5** | **1.5** |
>
> [1] Daiqing Qi, Handong Zhao, and Sheng Li. Better generative replay for continual federated learning. In The Eleventh International Conference on Learning Representations, ICLR 2023, Kigali, Rwanda, May 1-5, 2023. OpenReview.net, 2023.
>
> ---
>
>
> #### ***Question 3: Do task transitions for the clients happen simultaneously, or do tasks change in different clients independently?***
> ****Answer:**** Following the existing work of FCL, we assume task transitions for the clients happen simultaneously.
>
> ---

---

> ### Author Response · Authors · 2023-11-17
> **Response to Reviewer T35E (part 2)**
>
> #### ***Question 4: Is any example saved in the memory for AF-FCL? What is the memory size for the memory-based baselines (GFCL and FLwF2T)?***
> ****Answer:**** (1). **There is no example saved in memory for AF-FCL.** Following the setting in the original paper, we set the exemplar memory as shown in the table below.
>
> (2). We set the same size of exemplar memory for GFCL and FLwF2T. Besides, for fair comparison, we add perturbations to all prototype samples, as in GFCL.
>
> | | EMNIST | CIFAR100 | MNIST-SVHN-F |
> | --- | --- | --- | --- |
> | GFCL | 1000 | 2000 | 3600 |
> | FLwF2T | 1000 | 2000 | 3600 |
> | AF-FCL | 0 | 0 | 0 |
>
> ---
>
> #### ***Question 5: How robust is AF-FCL to data heterogeneity? Could you please include the information regarding how heterogeneous clients' data distribution is?***
> ****Answer:**** We would like to thank the reviewer for the helpful advice.
>
> 1. **By selectively utilizing global knowledge, AF-FCL is highly robust to statistical heterogeneity.** We accurately identify benign knowledge through probability estimation, thus mitigating the adverse impact of statistical heterogeneity and erroneous information.
>
>
> 2. **For the experiments in the paper, the statistical heterogeneity among clients is simulated by randomly assigning different classes of data to each client.** For example, in the EMNIST dataset, each client possesses 12 classes of data among all 26 classes. In the table below, we display the probability of any two different clients possessing the same class of data and the probability of any two tasks for two different clients possessing the same class of data. We calculate the probability using the actual class overlaps in the constructed datasets.
>
> | | two clients possessing the same class | two tasks possessing the same class|
> | --- | --- | --- |
> | EMNIST-LTP | 48% | 6% |
> | EMNIST-shuffle | 100% | 15% |
> | CIFAR100 | 78% | 16% |
> | MNIST-SVHN-F | 58% | 8% | -->
>
>
>
> 3. **To further examine the influence of statistical heterogeneity, we supplement experiments on the EMNIST dataset distributed with a Dirichlet distribution to represent real-world data distribution patterns and control heterogeneity.** We partition the dataset into 48 tasks with Dirichlet distributions $Dir_{48}(0.1)$ and $Dir_{48}(0.1)$ following the work in [4]. And we randomly assign 6 tasks to each of the 8 clients. The results of our method and baselines are shown in the table below. Under a more heterogeneous setting ($Dir_{48}(0.1)$), the collaboration among users has weakened, and the potential bias has intensified, resulting in a significant decline in the performance of all methods. While AF-FCL outperforms all baselines under both high and low degrees of heterogeneity by selectively utilizing learned knowledge.
>
> | Model | $Dir_{48}(0.1)$ | $Dir_{48}(0.1)$ | $Dir_{48}(0.5)$ | $Dir_{48}(0.5)$ |
> | --- |  --- |  --- |  --- |  --- |
> |  | Accuracy | Forgetting | Accuracy | Forgetting |
> | FedAvg | 47.3 | 13.4 | 63.9 | 8.7 |
> | FedProx | 47.6  | 13.7 | 63.8 | 8.2 |
> | PODNet+FedAvg | 51.0 | 10.7 | 63.9 | 8.3 |
> | PODNet+FedProx | 50.7 | 10.6 | 64.6 | 7.5 |
> | ACGAN-Replay+FedAvg | 52.4 | 9.1 | 66.7 | 5.2 |
> | ACGAN-Replay+FedProx | 54.7 | 8.3 | 66.0 | 5.9 |
> | FLwF2T | 50.2 | 11.8 | 66.3 | 5.0 |
> | FedCIL | 55.0 | 8.7 | 66.8 | 4.9 |
> | GLFC | 52.2 | 9.3 | 65.3 | 5.1 |
> | AF-FCL | **59.7** | **8.0** | **70.0** | 5.1 |
>
> ---
>
>
> #### ***Question 6: What is the communication cost for AF-FCL?***
> ****Answer:**** **The communication mode of AF-FCL is the same as FedAvg.** After training locally, the clients send gradients of the model to the server. Then the server transmits the aggregated gradients back to clients. Therefore, same as other baselines, the communication cost for AF-FCL is: number of clients $\times$ number of communication rounds $\times$ number of model parameters.
>
> ---
>
>
> #### ***Question 7: What is the number of parameters in NF models?***
> ****Answer:**** The number of parameters in NF models from AF-FCL is shown in the following table. In comparison, we also show the number of paraemters in GAN models from FedCIL.
>
> | | EMNIST | CIFAR100 |
> | --- | --- | --- |
> | FedCIL (GAN) | 4.993 M | 5.902 M |
> | AF-FCL (NF) | 5.943 M | 6.398 M |
>
> From the above table, the number of parameters in the NF models is comparable to the GAN models.
>
> ---
>
> #### ***Question 8: Are all the clients participating in the training every round?***
> ****Answer:**** Following the setting of existing works, all the clients participate in the training every round in both the proposed method and baselines.
>
> ---

---

> ### Author Response · Authors · 2023-11-17
> **Response to Reviewer T35E (part 3)**
>
> #### ***Question 9: How is g initialized? What is the forgetting in the NF model?***
> ****Answer:**** We would like to explain it as follows:
>
> 1. The parameters of the NF model are initialized randomly from a normal distribution. While learning in the first round of the first task, the NF model is only trained but not used.
> 2. The NF model evaluates the benefits of generating features for the current task through probabilistic estimation. It enables the classifier to forget erroneous features by assigning lower weights to biased features. Then, by learning from the feature space of the purified classifier, the NF model also forgets erroneous memories.

---

> ### Author Response · Authors · 2023-11-19
> **Need further Clarification?**
>
> Dear reviewer T35E:
>
> We would like to appreciate your precious efforts in reviewing our paper. Meanwhile, we have done our best to clarify all your concerns in our rebuttal. For example, for the issue you are most concerned about: **the number of clients is small in the experiments**, we would like to clarify that we followed existing works and have supplemented experiments, which we elaborate on in the answers to **Question 2**.
>
> Therefore, would you mind checking our response, and is there any unclear point so that we could further clarify?
>
> Best regards,
>
> Authors

---

> > ### Author Response · Authors · 2023-11-22
> > **As the discussion period is closing, is there a need for further clarification?**
> >
> > Dear reviewer T35E:
> >
> > We apologize for any inconvenience our request may cause during your schedule. As the rebuttal phase is drawing to a close, we would be grateful if you could review our responses. We have made every effort to address the concerns you raised thoroughly and thoughtfully. If you have any other questions, we hope you can communicate with us again. Your insights are crucial in helping us refine and improve our research.
> >
> > Best regards,
> >
> > Authors

---

> > > ### Comment · Reviewer_T35E · 2023-11-22
> > > **Answer to Rebuttal**
> > >
> > > I want to thank the authors for their comprehensive response. I do not have any further questions, and I have increased my score to 8.

---

> > > > ### Author Response · Authors · 2023-11-22
> > > > **Many thanks for raising the score**
> > > >
> > > > We express our sincere gratitude for your practical suggestions, which have provided us with valuable insights. The discussion regarding the difficulty of baseline methods has further deepened the analysis of the experimental results. And the supplementation of experiments with more clients can enhance the persuasiveness of our paper. We will adhere to these recommendations in the final version and also revise the paper according to all other comments.

---

### Official Review · Reviewer_BJsd · 2023-11-02

**Soundness:** 4 excellent
**Presentation:** 2 fair
**Contribution:** 3 good
**Rating:** 8
**Confidence:** 4

**Summary:**

This work suggested the harm of remembering biased or irrelevant features, which could happen in federated continue learning (FCL) scenarios, and designed a generative methods to mitigate erroneous information by correlation estimation with an NF model. The authors have conducted sufficient experiments to validate the effectiveness of the proposed method.

**Strengths:**

1. The methodology is solid, successfully excluding biased features from the memory bank.

2. The experiments are sufficient.

**Weaknesses:**

1. The problem formulation is confusing. In section 3.1 and most parts of section 3.2, the authors explain federated continual learning and a limitless task pool in detail, which is irrelevant to the methodology section. If I understand correctly, the main contribution of this work is disentangling and removing biased harmful features, while FCL merely serving as a relevant scenario. It would be better if the authors introduced the FCL formulation briefly and focused on the biased features in the memory bank.

2. It would be better if the authors could illustrate or formally define "biased features."

3. Despite EMNIST-noisy, the authors haven't explained the reason why using all other datasets could introduce biased features.

4. The term 'task' in the paper seems to refer to the 'data domain', instead of the commonly used task definitions (e.g., classification, segmentation, edge estimation), which is confusing.

**Questions:**

See weaknesses.

---

> ### Author Response · Authors · 2023-11-17
> **Response to Reviewer BJsd (part 1)**
>
> We would like to thank the reviewer for the positive and insightful comments. Below are our responses to the comments in **Weaknesses**.
>
> ---
>
> #### ***Question 1: The main contribution of this work is disentangling and removing biased harmful features, while FCL merely serving as a relevant scenario. It would be better if the authors introduced the FCL formulation briefly and focused on the biased features in the memory bank.***
>
> **Answer:** Thanks for the valuable comments. As you mentioned, our main contribution of this work is disentangling and removing biased harmful features. More importantly, we find that forgetting could be beneficial for model learning. FCL is a natural scenario where statistical heterogeneity exists, making it possible to improve performance by accurating forgetting. We agree that our method may be applicable in other scenarios besides FCL. Following the advice, we have revised our paper and discussed the practicability of our method in other learning scenarios.
>
> ---
>
> #### ***Question 2: It would be better if the authors could illustrate or formally define "biased features."***
> ****Answer:**** Thanks for the constructive advice. We have supplemented the definition of biased features in the paper.
>
> **Researchers have employed various definitions for biased features, one of which involves defining them as spurious correlations.** We denote $\mathcal{X}$, $\mathcal{Y}$ as the input and output spaces of machine learning algorithms. An algorithm learns a mapping from the data $x\in\mathcal{X}$ to the prediction $\hat{y}\in\mathcal{Y}$: $\hat{y}=f(x)$. We assume there are attributes $\gamma_1, \gamma_2,...$ abstracted from the data $x$. For example, $\gamma_1$ represents the shape of the object in the input image $x$, and $\gamma_2$ denotes the number of black pixels in the input image $x$. The machine learning algorithm actually relies on many attributes to infer: $\hat{y}=f(\gamma_{i_1}, \gamma_{i_2},..., \gamma_{i_N})$. We define an attribute $\gamma$ as a biased feature if it does not comply with the natural meaning of the target $y$ [1]. Relying on such a biased attribute would result in poor generalizability of the algorithm. The biased features could be attained through a biased training dataset, and the learned mapping $f$ relying on the biased features may not perform well in the testing dataset. For instance, if in the training image dataset all cows are standing on the grass, the machine learning model may rely on the attribute 'grass' for classifying images of cows.
>
> [1] Jeon, Myeongho, et al. A conservative approach for unbiased learning on unknown biases. Proceedings of the IEEE/CVF Conference on Computer Vision and Pattern Recognition. 2022.
>
> ---
>
>
> #### ***Question 3: Despite EMNIST-noisy, the authors haven't explained the reason why using all other datasets could introduce biased features.***
>
> ****Answer:**** The reasons other datasets could introduce biased features are as follows:
>
> 1. **There could be noisy data in all datasets, including images with gaussian noise or falsely labeled data.** These noisy data could result in biased features.
>
> 2. **Statistical heterogeneity could exacerbate the harmful effects of biased features.** The datasets in our study are distributed across various clients, exhibiting statistical heterogeneity. This heterogeneity, coupled with instances of insufficient or biased training data, can lead to the emergence of biased features. Recent research has indicated that such statistical heterogeneity can exacerbate the adverse effects of these biased features [2].
>
> 3. **Task heterogeneity could introduce biased features in most FL scenarios.** In all datasets, there are many tasks. And there could be some unrelated tasks that result in spurious correlations with other tasks. For instance, in the MNIST-SVHN-F dataset, different tasks rely on different features. Shape features that are relevant to digit classification in MNIST differ significantly from those that are important for classifying clothing items in FashionMNIST. If clients collaborate naively, it may result in a model that relies too heavily on spurious correlations, thus neglecting the significance of task-specific features.
>
> [2] Hong, Junyuan, et al. Federated robustness propagation: Sharing adversarial robustness in federated learning. (2021).
>
> ---

---

> > ### Author Response · Authors · 2023-11-17
> > **Response to Reviewer BJsd (part 2)**
> >
> > #### ***Question 4: The term 'task' in the paper seems to refer to the 'data domain', instead of the commonly used task definitions (e.g., classification, segmentation, edge estimation), which is confusing.***
> > ****Answer:**** Thanks for the valuable question. We would like to explain it as follows:
> >
> > We agree that in many Continual Learning (CL) settings, the meaning of 'task' closely resembles that of 'data domain'. In the scenarios of CL and FCL, a task corresponds to a labeled dataset (data domain). For example, in class incremental learning from CL, the classification of several new classes constitutes a task, as in our paper. **We follow the existing works of CL and FCL for consistency and use task to denote data domain.**

---

> > > ### Comment · Reviewer_BJsd · 2023-11-22
> > >
> > > Thanks for the response. My concerns have been resolved and I have raised my scores to 8.

---

> > > > ### Author Response · Authors · 2023-11-23
> > > > **Many thanks for raising the score**
> > > >
> > > > We greatly appreciate your pragmatic suggestions, which have significantly contributed to the enhancement of our paper's quality. Further clarification of biased features and a detailed exploration of bias in the experiments conducted are essential. Additionally, a clearer definition of the tasks would enhance the paper's coherence and rationale. We will integrate these comments into the final version and will also make revisions based on all other feedback received.

---

### Official Review · Reviewer_cH3G · 2023-11-03

**Soundness:** 3 good
**Presentation:** 3 good
**Contribution:** 2 fair
**Rating:** 5
**Confidence:** 2

**Summary:**

This paper presents Accurate Forgetting (AF) for Federated Continual Learning (FCL), a method that effectively leverages past knowledge in federated networks. It tackles the issue of biased or irrelevant features in FCL due to statistical heterogeneity. AF weights the generated data based on the current data distribution, effectively "forgetting" data that is heterogeneous to the current client and task. It also memorizes previous data by generating pseudo feature vectors based on the distribution in the latent space, thereby retaining information from previous data.

**Strengths:**

1. Rather than preventing forgetting in continual learning settings, the authors introduce an interesting concept that forgetting is crucial even in these settings. They propose a method that accurately forgets heterogeneous or malign information by assigning lower weights to certain generated feature vectors.
2. They employ a normalizing flow model to retain previous knowledge through distribution in feature space.
3. The results, supported by ablation studies, highlight the effectiveness and superiority of their proposed accurate forgetting method over existing leading-edge methods.

**Weaknesses:**

1. In Section 4.2's experiment, noise interference is limited to the initial three tasks. This doesn't adequately assess the impact on each method when the noise occurs at random or during intermediate stages, which would be a more general scenario.

2. The problems in Section 4.2 appear to overlap with those discussed in studies on Concept Drifts in Federated Learning (FL). To emphasize the importance and influence of this research, it would be beneficial to distinguish it from Concept Drifts in Federated Learning, incorporate these into the related works, or conduct additional experiments. Consider citing works such as FEDDRIFT (Jothimurugesan et al., 2022), ADAPTIVEFEDAVG (Canonaco et al., 2021), and Flash (Panchal et al., 2023).

3. AF-FCL allows each client to create a feature space distribution, incorporating information from all clients from the previous round. This potential problem is not found in standard FL methods.

**Questions:**

1. In section 5.3, the author claims that "learning in feature space avoids the danger of leaking raw data through the generative model, thus protecting data privacy". However, a client could theoretically generate raw data by first using the NF model from the previous FedAvg to generate numerous features, then converting these back to the data space using deconvolution methods with h_a from the last FedAvg.
Even assuming deconvolution is difficult, AF-FCL still permits each client to generate a feature space distribution, which includes information from all clients in the previous round. This potential problem is not found in standard FL methods.
Is there a more effective solution to this data privacy concern?

2. The potential for malicious data to emerge during the Federated task is a concern. The proposed AF-FCL appears to "accurately forget" prior correct data, an issue not expected with other FCL methods that aim at memorizing everything. The effects are unclear in the following scenarios:
*    The malicious (falsely labeled) data occurs in the middle of the FL task.
*    Increased heterogeneity among clients and tasks during the FL task, and comparison of AF-FCL to the baseline.

3. Please provide a more plausible real-world scenario where various classification tasks need to be trained within the same model under Federated Learning (FL). This would illustrate the necessity for "accurate forgetting" to manage high levels of heterogeneity. Figure 1 provides a clear example. However, if each hospital has different classification tasks (thorax, brain, liver) that they wish to train together under FL, wouldn't it be more reasonable to conduct them in three separate FL environments?

---

> ### Author Response · Authors · 2023-11-17
> **Response to Reviewer cH3G (part 1)**
>
> We would like to thank the reviewer for the insightful comments. Below are our responses to the comments in **Weaknesses** and **Questions**.
>
> ---
>
> #### ***Question 1: In Section 4.2's experiment, noise interference is limited to the initial three tasks. This doesn't adequately assess the impact on each method when the noise occurs at random or during intermediate stages, which would be a more general scenario.***
> ****Answer:**** Thanks for the insightful comments. We would like to clarify it as follows:
>
> 1. **In fact, our method is naturally suitable for cases where the noise occurs at random or during intermediate stages.** In this paper, we address the scenario where biased information resides in the memory bank and propose methods to mitigate its detrimental impact on the current task. While our method can also manage the situation where noisy tasks occur at random stages or during intermediate stages. When the model learns on the current task containing noise, AF-FCL measures the probability $p_D(h_{a}(x_i))$ of **local data $x_i$**, rather than the generated data, within the global data distribution using the NF model. And the loss objective training on local data is as follows:
>
> $$
> \mathcal{L}_{ce}^x(h; D_k^t)=\frac{1}{n_k^t}\sum\_{i=1}^{n\_k^t}k\cdot p\_D(h\_{a}(x\_i))\cdot\mathcal{L}\_{ce}(h(x\_i), y\_i),
> $$
>
> where $k\cdot p_D(h_{a}(x_i))$ denotes the weights of the local data in the loss objective $\mathcal{L}_{ce}^x$ and $k$ is a hyper-parameter. The noise in the current task typically has  relatively lower probability compared with the beneficial feature. By using the global probability as the weight in the loss objective, we mitigate the adverse effects of low-probability biased data of current task.
>
> 2. **We experimentally validate that AF-FCL could address the noise when it occurs at random or during intermediate stages.** We supplement experiments when the noisy tasks in Section 4.2's experiment occur at random stages. The average accuracy and forgetting of normal tasks are displayed in the table below. Our method consistently surpasses all baselines by alleviating the negative influence of noisy clients.
>
> | Model | $M=1$ |  $M=1$ |  $M=2$ | $M=2$ |
> | --- | --- | --- | --- | --- |
> | | Accuracy | Forgetting | Accuracy | Forgetting |
> | FedAvg | 48.7 | 22.5 | 46.5 | 23.6 |
> | FedProx | 48.5 | 21.8 | 47.8 | 21.5 |
> | PODNet+FedAvg | 36.3 | 30.7 | 28.7 | 28.2 |
> | PODNet+FedProx | 34.7 | 28.9 | 29.9 | 31.8 |
> | ACGAN-Replay+FedAvg | 35.4 | 27.6 | 30.9 | 26.4 |
> | ACGAN-Replay+FedProx | 39.9 | 26.4 | 35.7 | 26.3 |
> | FLwF2T | 43.2 | 22.4 | 39.5 | 25.1 |
> | FedCIL | 45.7 | 24.1 | 40.6 | 26.8 |
> | AF-FCL | **50.0** | **20.6** | **48.2** | **21.4** |
>
>
> ---
>
>
> #### ***Question 2: The problems in Section 4.2 appear to overlap with those discussed in studies on Concept Drifts in Federated Learning (FL). To emphasize the importance and influence of this research, it would be beneficial to distinguish it from Concept Drifts in Federated Learning, incorporate these into the related works, or conduct additional experiments.***
>
> ****Answer:**** Thanks for the insightful advice about expanding the horizons of this research. Following your kind advice, we have supplemented the discussion and cited the related work regarding the studies on Concept Drifts in Federated Learning.
>
> Different from the studies about Federated Continual Learning, the evaluation in the concept drift studies is conducted at each time step. Therefore, **there is no memorization requirement or catastrophic forgetting problem in the concept drift studies.** [1] proposed a novel clustering algorithm for reacting to concept drifts. Adaptive-FedAVG adapted the learning rate to react to concept drift [2]. [3] proposed to detect concept drift through the magnitude of parameter updates and designed a novel adaptive optimizer.
>
> [1] Ellango Jothimurugesan, Kevin Hsieh, Jianyu Wang, Gauri Joshi and Phillip B. Gibbons. Federated Learning under Distributed Concept Drift. International Conference on Artificial Intelligence and Statistics, 25-27 April 2023, Palau de Congressos, Valencia, Spain.
>
> [2] Canonaco, Giuseppe, et al. Adaptive federated learning in presence of concept drift. 2021 International Joint Conference on Neural Networks (IJCNN). IEEE, 2021.
>
> [3] Panchal, Kunjal, et al. Flash: Concept Drift Adaptation in Federated Learning. 2023.
>
> ---

---

> ### Author Response · Authors · 2023-11-17
> **Response to Reviewer cH3G (part 2)**
>
> #### ***Question 3: A client could theoretically generate raw data by first using the NF model from the previous FedAvg to generate numerous features, then converting these back to the data space using deconvolution methods. AF-FCL permits each client to generate a feature space distribution, which includes information from all clients in the previous round. This potential problem is not found in standard FL methods. Is there a more effective solution to this data privacy concern?***
> ****Answer:**** Thanks for pointing out the issue. We would like to clarify it as follows:
>
> 1.  **Compared with related works in the FCL domain, our approach incorporates more considerations regarding privacy.** For instance, a global generative model for raw data is applied in [4]. Compared with baselines, AF-FCL generates features rather than raw data, which mitigates privacy leakage. And we only transmit gradients of model parameters between clients and servers, as in FedAvg. **Besides, there are several works in FL utilizing generative models in the feature space of classifiers to aid collaboration among clients [5,6].**
>
> 2. **Our method can be easily modified to provide enhanced privacy protection and avoid the leakage of raw data through deconvolution methods.** Instead of sharing the complete classifier among clients, we can keep the feature extrator of the classifier personalized and local for each client, only sharing the classifier head and NF model between the server and clients. By concealing the local feature extrator, which is the only model processing raw data, we can better protect client privacy. Without access to the feature extrator, one cannot attain raw data by deconvolution methods.
>
> 3. **Our method is also compatible with existing privacy protection techniques like differential privacy.** By elaborately adding noise to the local model updates and homomorphic encryption during model aggregation, we can train a global model adhering to a predefined privacy budget.
>
> [4] Daiqing Qi, Handong Zhao, and Sheng Li. Better generative replay for continual federated learning. In The Eleventh International Conference on Learning Representations, ICLR 2023, Kigali, Rwanda, May 1-5, 2023. OpenReview.net, 2023.
>
> [5] Zhu, Zhuangdi, Junyuan Hong, and Jiayu Zhou. Data-free knowledge distillation for heterogeneous federated learning. International conference on machine learning. PMLR, 2021.
>
> [6] Wu, Yuezhou, et al. Fedcg: Leverage conditional gan for protecting privacy and maintaining competitive performance in federated learning. arXiv preprint arXiv:2111.08211 (2021).
>
> ---
>
>
> #### ***Question 4: The proposed AF-FCL appears to "accurately forget" prior correct data. The effects are unclear in the following scenarios: (a) The malicious (falsely labeled) data occurs in the middle of the FL task. (b) Increased heterogeneity among clients and tasks during the FL task, and comparison of AF-FCL to the baseline.***
> ****Answer:**** Thanks for the very constructive comments. We would like to clarify it as follows:
>
> 1. **For scenario (a), we guess this problem is similar to Question 1.** Please refer to the answer to **Question 1**. If we misunderstand this problem, please feel free to point it out at any time.
>
> 2. **For scenrio(b), we experimentally validated that AF-FCL could effectively utilize memorized knowledge in scenarios with increased heterogeneity.**  We supplement experiments on the EMNIST dataset distributed with a Dirichlet distribution to represent real-world data distribution patterns and control heterogeneity. We partition the dataset into 48 tasks with Dirichlet distribution $Dir_{48}(0.1)$ and $Dir_{48}(0.1)$ following the work in [4]. And we randomly assign 6 tasks to each of the 8 clients. The results of our method and baselines are shown in the table below. Under a less heterogeneous setting ($Dir_{48}(0.5)$), clients collaborate more closely with each other, thus the accuracy is higher. From the table below, AF-FCL outperforms all baselines under both high and low degrees of heterogeneity.
>
> | Model | $Dir_{48}(0.1)$ | $Dir_{48}(0.1)$ | $Dir_{48}(0.5)$ | $Dir_{48}(0.5)$ |
> | --- |  --- |  --- |  --- |  --- |
> |  | Accuracy | Forgetting | Accuracy | Forgetting |
> | FedAvg | 47.3 | 13.4 | 63.9 | 8.7 |
> | FedProx | 47.6  | 13.7 | 63.8 | 8.2 |
> | PODNet+FedAvg | 51.0 | 10.7 | 63.9 | 8.3 |
> | PODNet+FedProx | 50.7 | 10.6 | 64.6 | 7.5 |
> | ACGAN-Replay+FedAvg | 52.4 | 9.1 | 66.7 | 5.2 |
> | ACGAN-Replay+FedProx | 54.7 | 8.3 | 66.0 | 5.9 |
> | FLwF2T | 50.2 | 11.8 | 66.3 | 5.0 |
> | FedCIL | 55.0 | 8.7 | 66.8 | 4.9 |
> | GLFC | 52.2 | 9.3 | 65.3 | 5.1 |
> | AF-FCL | **59.7** | **8.0** | **70.0** | 5.1 |
>
> [4] Wang, Jianyu, et al. Tackling the objective inconsistency problem in heterogeneous federated optimization. Advances in neural information processing systems 33 (2020): 7611-7623.
>
> ---

---

> > ### Author Response · Authors · 2023-11-17
> > **Response to Reviewer cH3G (part 3)**
> >
> > #### ***Question 5: Please provide a more plausible real-world scenario where various classification tasks need to be trained within the same model under Federated Learning (FL). Figure 1 provides a clear example. However, if each hospital has different classification tasks (thorax, brain, liver) that they wish to train together under FL, wouldn't it be more reasonable to conduct them in three separate FL environments?***
> > ****Answer:**** Thanks for the inspiring question. We would like to explain it as follows:
> >
> > 1. **A more plausible real-world scenario where various classification tasks need to be trained within the same model under Federated Learning (FL) is a cluster of autonomous learning robots.** Each robot gradually acquires knowledge about the world. It learns new tasks while retaining knowledge from previous tasks, such as initially learning simple color classification and then progressing to animal classification. Moreover, each robot can leverage the knowledge acquired by other robots. Due to the correlation among the progressing tasks, various classification tasks could be trained within the same model. And due to the benefits of collaboration among robots and data privacy constraints, they can be trained under federated learning.
> >
> > 2. **In Figure 1, considering the correlation between diseases, learning a shared model may enhance performance [5].**  For example, the association between cardiovascular (heart-related) and cerebrovascular (brain-related) diseases suggests that learning these tasks within a unified model could capture their interrelations, potentially benefiting the learning process of both. Thanks for pointing out the issue; we have supplemented the clarification of the correlation among diseases in the paper.
> >
> > [5] Dong, Jiahua, et al. Federated class-incremental learning. Proceedings of the IEEE/CVF conference on computer vision and pattern recognition. 2022.

---

> ### Author Response · Authors · 2023-11-19
> **Need further Clarification?**
>
> Dear reviewer cH3G:
>
> We would like to appreciate your precious efforts in reviewing our paper. Meanwhile, we have done our best to clarify all your concerns in our rebuttal. For example, for the issue you are most concerned about: **situations where noise occurs at random stages**, we would like to clarify that our framework is naturally capable of dealing with such situations, which we elaborate on in the answers to **Question 1** and **Question 4**.
>
> Therefore, would you mind checking our response, and is there any unclear point so that we could further clarify?
>
> Best regards,
>
> Authors

---

> ### Author Response · Authors · 2023-11-22
> **As the discussion period is closing, is there a need for further clarification?**
>
> Dear reviewer cH3G:
>
> We apologize for any inconvenience our request may cause during your schedule. As the rebuttal phase is drawing to a close, we would be grateful if you could review our responses. We have made every effort to address the concerns you raised thoroughly and thoughtfully. If you have any other questions, we hope you can communicate with us again. Your insights are crucial in helping us refine and improve our research.
>
> Best regards,
>
> Authors

---

### Official Review · Reviewer_pq5J · 2023-11-03

**Soundness:** 3 good
**Presentation:** 3 good
**Contribution:** 3 good
**Rating:** 8
**Confidence:** 4

**Summary:**

The authors consider the problem of Federated Continual Learning (FCL) in which each client is faced with a sequence of (potentially unrelated) learning tasks. The challenge to be solved is that clients maintain performance across all tasks seen so far while learning in a collaborative fashion s.t. they learn a global model capable of handling all tasks seen by all clients during training.
A novel method is proposed to tackle a rather general definition of FCL leveraging generative replay with Normalizing Flows (NF), Knowledge Distillation to control feature distribution shift and accurate forgetting which aims to exclude uninformative/harmful samples/features from learning. Experiments were conducted on multiple datasets to show the effectiveness of the proposed method.

**Strengths:**

- the FCL definition used here allows for a broad set of continual learning problems in FL scenarios, including scenarios with unrelated or even contradictory tasks (termed Limtless Task Pool (LTP))
- the problem is relevant, interesting and well motivated (especially in Sec. 4)
- the paper is written clearly in general. The problem definition as well as the methodogical contribution is clearly elaborated on
- it is straightforward to follow through the paper
- experiments show the effectiveness of the approach and an ablation study gives evidence that the proposed method works well in the presented setting
- code is available

**Weaknesses:**

# Section 4
While the problem shown in Sec. 4 sufficiently supports the parts of the motivation of the paper, it lacks to clearly support the claim that biases in the data have severe impact on the model performance in FCL. However, related work already indicated that this is the case, making an explicit empirical validation somewhat obsolete. Nevertheless it would be great to have some references to fully cover the claims in this section as well.

# Section 5
- "Besides, learning in feature space avoids the danger of leaking raw data through generative model, thus protecting data privacy". This does not protect privacy if the model is leaked because one could still run e.g. membership inference attacks given the first feature extractor. Without any additional protection mechanism (e.g. differential privacy) privacy guarantees are not given. Please rephrase that part. [1][2]

- "The invertability ensures a lossless memory of the original input." Yes, but it's more the bijective property that allows for a lossless memory since if the transformations were not bijective, one could not reconstruct the output uniquely. Please consider rephrasing.

- regarding Correlation Estimation: does a large distance of encodings in the latent space of the NF  (i.e. low likelihood)  imply that tasks/samples/features are contradictory? I think it does not, although your proposed way to measure this might have a certain level of accuracy since the distance and the extent of being contradictory probably correlate. It would be beneficial to see how strong the correlation between distance of tasks in the latent space and "harmfulness" during learning is.

# Section 6
- although the experiments cover most of the claims in the paper, there are still doubts to which extent the empirical results support the claims: All experiments consider image classification tasks of either the same dataset (split into several tasks) or very related datasets (e.g. MNIST and SVHN). Although in the LTP (using EMNIST) setting tasks are sampled randomly by each client, there still might be a quite high correlation among tasks as images of different letters still may share certain features (e.g. "b" and "p"). An additional experiment on a more diverse datasaet such as CIFAR-100 in the LTP setting would be beneficial to see.

# Minor points
Please be consistent in the notation:
- in the CL (Sec.3) definition $\mathcal{T}^i$ and $D^i$ both refer to datasets.
- when introducing CL and FL in Sec. 3, in the CL definition the indexes change: While in the CL definition $x^t_i$ refers to the $i$-th sample of task $t$, $x^i_k$ refers to the $i$-the sample on the $k$-th client in the FL definition.
- Eq. 1: the last loss should be $\mathcal{L}(\theta^t, \mathcal{T}^T)$

# References
[1] Suri et. al. 2022. Subject Membership Inference Attacks in Federated Learning.

[2] Hatamizadeh et. al. 2022. Do Gradient Inversion Attacks Make Federated Learning Unsafe?

**Questions:**

# Section 5
- Assume a task $T_{t-1}$ has been learned successfully and now a client faces task $T_t$ which is unrelated to $T_{t-1}$ but $T_{t-1}$ is not contradictory to $T_t$, i.e. does not harm learning. Since the tasks are unrealted it is likely that they are far off each other in the latent space of the NF, hence the proposed method would weight down $T_{t-1}$ while learning $T_t$. Forgetting in this case would be harmful in this case. Have you considered such cases in the experiments?

---

> ### Author Response · Authors · 2023-11-17
> **Response to Reviewer pq5J (part 1)**
>
> We would like to thank the reviewer for the positive and very valuable comments. Below are our responses to the comments in **Weaknesses** and **Questions**.
>
> ---
>
> #### ***Question 1: In Section 4, it lacks to clearly support the claim that biases in the data have severe impact on the model performance in FCL. Related work already indicated that this is the case. It would be great to have some references to fully cover the claims in this section as well.***
>
>
> ****Answer:**** Thanks for the valuable advice.
> 1. **As the reviewer mentioned, we concur that discussion regarding the impact of bias on the model performance in FCL, supported by references, would indeed enhance the manuscript's rigor and depth.** For instance, [1] demonstrated that statistical heterogeneity in FCL results in severe performance degradation. [2] theoretically and empirically revealed that bias could propagate and be exaggerated through federated learning. [3] designed a synthetic biased dataset and found that it critically reduced the performance of continual learning methods.
>
> 2. **Besides, we supplement experiments on the EMNIST-noisy dataset without malicious clients ($M=0$) to study the impact of biases.** The results of the last 3 tasks in the EMNIST-noisy dataset are shown in the Table below. The dataset has an increasing number of malicious clients, denoted as $M$. The efficacy of the methods declines with more malicious clients. Furthermore, introducing bias into the dataset (from $M=0$ to $M=1$) significantly undermines the performance of both the baselines and the proposed method.
>
> | Model | $M=0$ | $M=0$ | $M=1$ | $M=1$ | $M=2$ | $M=2$ | $M=4$ | $M=4$ |
> | --- | --- | --- | --- | --- | --- | --- | --- | --- |
> | | Accuracy | Forgetting | Accuracy | Forgetting | Accuracy | Forgetting | Accuracy | Forgetting |
> | FedAvg | 52.6 | 17.1 | 52.3 | 16.1 | 51.7 | 16.0 | 50.4 | 12.1 |
> | FedProx | 53.4 | 12.6 | 52.5 | 12.5 | 51.8 | 18.8 | 51.0 | 13.5 |
> | PODNet+FedAvg | 53.3 | 24.8 | 43.3 | 20.3 | 38.5 | 20.1 | 33.8 | 19.0 |
> | PODNet+FedProx | 54.3 | 23.9 | 44.3 | 19.6 | 37.3 | 21.2 | 34.1 | 18.4 |
> | ACGAN-Replay+FedAvg | 58.4 | 11.9 | 45.8 | 18.6 | 42.6 | 17.5 | 40.2 | 16.0 |
> | ACGAN-Replay+FedProx | 60.4 | 6.7 | 50.2 | 18.5 | 43.7 | 17.2 | 39.6 | 16.4 |
> | FLwF2T | 65.9 | 4.9 | 52.1 | 14.7 | 47.6 | 18.6 | 44.5 | 14.1 |
> | FedCIL | 66.5 | 5.6 | 49.8 | 15.2 | 45.8 | 19.1 | 42.0 | 15.8 |
> | AF-FCL | **73.7** | **1.7** | **55.5** | **7.5** | **54.9** | **11.8** | **54.0** | **12.8** |
>
> [1] Daiqing Qi, Handong Zhao, and Sheng Li. Better generative replay for continual federated learning. In The Eleventh International Conference on Learning Representations, ICLR 2023, Kigali, Rwanda, May 1-5, 2023. OpenReview.net, 2023.
>
> [2] Hongyan Chang, Reza Shokri. Bias Propagation in Federated Learning. In The Eleventh International Conference on Learning Representations, ICLR 2023, Kigali, Rwanda, May 1-5, 2023.
>
> [3] Lee, Donggyu, Sangwon Jung, and Taesup Moon. Issues for Continual Learning in the Presence of Dataset Bias. AAAI Bridge Program on Continual Causality. PMLR, 2023.
>
> ---
>
> #### ***Question 2: In Section 5, learning in feature space does not protect privacy if the model is leaked because one could still run e.g. membership inference attacks given the first feature extractor. Without any additional protection mechanism (e.g. differential privacy) privacy guarantees are not given. Please rephrase that part.***
>
>
> Thanks for pointing out the issue. We would like to explain it as follows:
>
> 1. **Indeed, we concur that learning in feature space does not guarantee absolute privacy protection. Accordingly, we have rephrased the statement.**
>
> 2. **We mentioned privacy protection because our approach incorporates more considerations regarding privacy compared with existing works in the FCL domain.** For instance, [1] proposed learning a global generative model for raw data to mitigate catastrophic forgetting in FCL.
>
>
> 3. **Our method can be easily modified to provide enhanced privacy protection and avoid membership inference attacks.** Instead of sharing the complete classifier among clients, we can keep the feature extrator of the classifier personalized and local for each client, only sharing the classifier head and NF model between the server and clients. By concealing the local feature extrator, which is the only model processing raw data, we can better protect client privacy. Without access to the feature extrator, membership inference attacks cannot be implemented.
>
> [1] Daiqing Qi, Handong Zhao, and Sheng Li. Better generative replay for continual federated learning. In The Eleventh International Conference on Learning Representations, ICLR 2023, Kigali, Rwanda, May 1-5, 2023. OpenReview.net, 2023.
>
> ---

---

> > ### Comment · Reviewer_pq5J · 2023-11-20
> > **Answer to Rebuttal**
> >
> > Thank you for your detailed answers!
> >
> > Most of my concerns were addressed & resolved, I increase my score to 8.
> >
> > However, regarding point 3 in Question 2, I don't agree: Since you still send the NF between servers and clients, an adversary still has access to sensible information encoded in the NF's distribution. You're right that e.g. membership inference attacks might not be possible (at least not that easily), however it does not increase privacy as private information is still captured in the NF.

---

> > > ### Author Response · Authors · 2023-11-22
> > > **Many thanks for raising the score**
> > >
> > > Thank you very much for your insightful suggestions, which have been greatly enlightening and are crucial for enhancing the quality of our paper! The suggestions regarding how the model processes irrelevant information are very enlightening. Similarly, conducting experiments on a more diverse set of datasets can enhance the persuasiveness of our paper. We will adhere to these recommendations in the final version and also revise the paper according to all other comments.

---

> ### Author Response · Authors · 2023-11-17
> **Response to Reviewer pq5J (part 2)**
>
> #### ***Question 3: In Section 5, it's more the bijective property than the invertibility that allows for a lossless memory of the NF model. Please consider rephrasing.***
> ****Answer:**** Thanks for the insightful comments. We have rephrased the statements. Although an invertible function is inherently bijective, it is specifically the bijective property that guarantees the lossless memory characteristic of the NF model.
>
> ---
>
> #### ***Question 4: In Section 5, does a large distance of encodings in the latent space of the NF (i.e. low likelihood) imply that tasks/samples/features are contradictory? I think it does not, although your proposed way to measure this might have a certain level of accuracy since the distance and the extent of being contradictory probably correlate. It would be beneficial to see how strong the correlation between distance of tasks in the latent space and "harmfulness" during learning is.***
> ****Answer:**** Thanks for the very constructive comments. We would like to explain it as follows:
>
> 1. **As the reviewer mentioned, the distance and the extent of being contradictory probably correlate.** Statistically, the method could help mitigate the undermining of malicious memory. We perform probability estimation of generated data with respect to local distribution to measure the relevance of the generated feature to current tasks. Outlier features could potentially be detrimental features. For example, we assume the NF model has memorized the mislabeled feature of label $Y$ from previous malicious tasks. While learning the correct task from data of label $Y$, the probability of memorized malicious features would be low since they are widely divergent from the correct feature with the same label $Y$.
>
> 2. **As the reviewer stated in Question 6, memorized knowledge from unrelated tasks may also present a large distance of encodings in the latent space, but it may not imply contradictory.** The impact of irrelevant knowledge and malicious knowledge on model training differs. Irrelevant knowledge, which does not contradict correct knowledge, can be retained during the acquisition of new tasks. For a more detailed explanation, please refer to the response provided in **Question 6**.
>
> 3. **Following the advice, we supplement the statistics of the measured weights for malicious features, relevant features, and irrelevant features in the following table.** On the EMNIST-noisy dataset in Section 4, 'malicious features' refer to the features whose label are the same with the subject features but are in the malicious clients; 'relevant features' refer to the features whose labels are the same with the subject features and are in the normal clients; 'irrelevant features' refer to the features whose labels are different from the subject features and are in the normal clients. These three kinds of features represent biased memory, benign memory, and irrelevant memory. The average probabilities of the three kinds in normal clients calculated by the NF model—in other words, the distance of encodings in the latent space of the NF model—are shown in the Table below. Relevant features present high probabilities in each client, and malicious features have low probabilities. Therefore, the negative impact of biased memory could be mitigated by assigning low weight to it. The probabilities of irrelevant features are lower than relevant features and higher than malicious features. Despite the somewhat low weight, the irrelevant features would not be forgotten by the model, as they do not conflict with the correct features, which we elaborate on in the answer of **Question 6**.
>
> | client | malicious features | relevant features | irrelevant features |
> | --- | --- | --- | --- |
> | client 1 | 0.1790 | 0.5791 | 0.3193 |
> | client 3 | 0.2375 | 0.7511 | 0.3320 |
> | client 4 | 0.0997 | 0.5948 | 0.3257 |
> | client 6 | 0.1358 | 0.5995 | 0.3081 |
> | client 7 | 0.1844 | 0.6789 | 0.2781 |
> | client 8 | 0.0528 | 0.6274 | 0.1548 |
>
> ---

---

> > ### Author Response · Authors · 2023-11-17
> > **Response to Reviewer pq5J (part 3)**
> >
> > #### ***Question 5: In Section 6, all experiments consider image classification tasks of either the same dataset (split into several tasks) or very related datasets (e.g. MNIST and SVHN). An additional experiment on a more diverse datasaet such as CIFAR-100 in the LTP setting would be beneficial to see.***
> > ****Answer:**** Many thanks for the insightful suggestions. We would like to explain it from two aspects:
> >
> > 1. **Following the advice, we have supplemented experiments on the CIFAR100 dataset with a more challenging LTP setting.** By randomly sampling 10 classes as a task among the 100 classes of CIFAR100, we construct 6 tasks for each of the 8 clients. The results in the Table below demonstrate that our method achieves superior accuracy relative to baselines. While CL methods and traditional FCL approaches focus on preserving previously acquired knowledge, they often inadvertently retain inaccurate information, adversely affecting performance on preceding tasks. Conversely, our method implements an adaptive strategy for selectively forgetting biased features. Therefore, in such a challenging setting with high statistical heterogeneity, this approach significantly diminishes forgetting, outperforming existing baselines, and thereby enhancing the retention of task-specific knowledge.
> >
> > | Model | Accuracy | Forgetting |
> > | --- |  --- |  --- |
> > | FedAvg | 19.5 | 2.4 |
> > | FedProx | 20.1 | 1.9 |
> > | PODNet+FedAvg | 21.3 | 2.0 |
> > | PODNet+FedProx | 21.6 | 2.1 |
> > | ACGAN-Replay+FedAvg | 19.5 | 3.0 |
> > | ACGAN-Replay+FedProx | 19.6 | 2.8 |
> > | FLwF2T | 21.5 | 5.9 |
> > | FedCIL | 19.6 | 2.9 |
> > | GLFC | 19.9 | 3.2 |
> > | AF-FCL | 23.8 | 0.9 |
> >
> >
> > 2. **In the MNIST-SVHN-F dataset, the tasks of the FashionMNIST dataset are designed for clothing image classification, different from digit classification tasks.** Different tasks rely on different features in this mixed MNIST-SVHN-F dataset. For example, shape features that are relevant to digit classification differ significantly from those that are important for classifying clothing items. If clients collaborate naively, it may result in a model that relies too heavily on spurious correlations, thus neglecting the significance of task-specific features. And the results on the MNIST-SVHN-F dataset demonstrate the superiority of our method.
> >
> >
> >
> > ---
> >
> > #### ***Question 6: When the newly arrived task is unrelated to old tasks but not contradictory to them, the proposed method would weight down old tasks while learning new tasks. Forgetting in this case would be harmful in this case. Have you considered such cases in the experiments?***
> > ****Answer:**** Many thanks for the profound question. **Though irrelevant features and harmful features are assigned lower weights during training, their impact on model training differs.**  The detailed explanation is as follows:
> > - ***The distinction between harmful and irrelevant features in relation to the current task is noteworthy. Specifically, classifiers derived from harmful features exhibit conflict with those of the current task, whereas irrelevant features do not present such conflict.*** For instance, if the NF model generates harmful features that misclassify circular images as squares and the current task is to correctly classify square images, this creates a conflict, making it hard to learn a consistent classifier. In contrast, irrelevant features, such as correctly classifying triangular images, can coexist within the same classifier that classifies squares.
> > - Returning to our training process, when the NF model generates harmful features, they are assigned a low probability estimate, prompting the classification model to prioritize the current task. Then, because the mappings of the current task and harmful memories conflict with each other, the classifier would learn the prioritized current mappings while forgetting those associated with harmful memories. In parallel, the NF model, by learning from the classifier's feature space, also progressively forgets these harmful memories. ***Conversely, when the NF model generates irrelevant memories, even though their estimated probability is also low, they do not conflict with the current task.*** As a result, the classification model, while concentrating on the current task, does not override these irrelevant memories, allowing their preservation through knowledge distillation and the NF model.

---

### Meta-Review · Area_Chair_knXr · 2023-12-05

**Metareview:**

The paper tackles a challenging and only recently sparsely explored topic of federated continual learning. Specifically, it proposes a normalizing flow based generative replay strategy to handle interference/forgetting in federated setups with highly heterogenous data.

The initial reviews for this paper were rather borderline, but a large portion of the reviewers' concerns could be resolved in the discussion phase and the ratings were raised correspondingly. Among many aspects, the raised points revolved in principle around clearer descriptions of settings, investigation of various baselines to ensure that the made statements are better supported, rationale behind the choice of normalizing flows, communication cost and privacy discussion, as well as more extensive evaluation.

In the discussion, the authors have added multiple descriptions on experimental tables to provide additional evidence and clarify. Some select points are left unaddressed, such as the claim that privacy is preserved and a justification behind the normalizing flow choice. Overall, these are easily addressable in a revision and should be included appropriately.

Given the discussion my current recommendation is to accept the paper, however, **I strongly emphasize that the paper is required to make the actual suggested and written changes**. As of now, the ability to upload a revised pdf has not been taken advantage of, despite the fact that the changes are numerous in nature. As reviewer hHgp also suggests, including the actual changes and adding the experimental parts are essential for this paper to be accepted.

**Justification For Why Not Higher Score:**

Although it seems like the majority of reviewers have raised their ratings due to comprehensive responses by the authors, the actual changes still need to be made to the paper and are numerous in nature. It will be a balancing act to include all the revisions into the pdf and a lot of effort is still required to polish the final version.

**Justification For Why Not Lower Score:**

Despite the many changes to still be made to the pdf, the provided responses by the authors are comprehensive enough that it is conceivable that a final revision will actually encompass the outlined improvements. In particular, the experimental results have in very large parts already been commented in the form of formatted tables, that only need to find their way into the pdf, suggesting that an update is highly likely to occur.

---

### Decision · Program_Chairs · 2024-01-16

Accept (poster)